# Pu.1/Spi1 dosage controls the turnover and maintenance of microglia in zebrafish and mammals

Yi Wu[1†], Weilin Guo[2†], Haoyue Kuang[3], Xiaohai Wu[3], Thi Huong Trinh[2], Yuexin Wang[3], Shizheng Zhao[1], Zilong Wen[2,3], Tao Yu[1,3]*

[1]Biomedical Research Institute, Shenzhen Peking University-the Hong Kong University of Science and Technology Medical Center, Shenzhen, China; [2]Division of Life Science, The Hong Kong University of Science and Technology, Hong Kong, China; [3]Greater Bay Biomedical Innocenter, Shenzhen Bay Laboratory, Shenzhen, China

## eLife Assessment

This study presents **valuable** findings on the regulation of survival and maintenance of brain-resident immune cells called microglia. Using **compelling** and sophisticated genetic tools, the authors demonstrate a gene dosage-dependent mechanism using which microglia are eliminated. This research on cell competition and survival will be of broad interest to the cell biology community.

*For correspondence:
tyu@connect.ust.hk

†These authors contributed equally to this work

Competing interest: The authors declare that no competing interests exist.

**Abstract** Microglia are brain-resident macrophages playing pivotal roles in central nervous system (CNS) development and homeostasis. Yet, the cellular and molecular basis governing microglia maintenance remains largely unknown. Here, by utilizing a visible conditional knockout allele of *pu.1/spi1b* gene (the master regulator for microglia/macrophage lineage development) to generate mosaic microglia populations in adult zebrafish, we show that while *pu.1*-deficient microglia are immediately viable, they are less competitive, and chronically eliminated through Tp53-mediated cell competition. Interestingly, when conditionally inactivating Pu.1 in adult *spi-b* (the orthologue of mouse *Spi-b*) null mutants, microglia are rapidly depleted via apoptosis, suggesting that Pu.1 and Spi-b regulate microglia maintenance in a dosage-dependent manner. The dosage-dependent regulation of microglia maintenance by PU.1/SPI1 is evolutionarily conserved in mice, as shown by conditionally inactivating single and both *Spi1* alleles in microglia, respectively. Collectively, our study reveals the conserved cellular and molecular mechanisms controlling microglia turnover and maintenance in teleosts and mammals.

## Introduction

Microglia, the crucial immune cells residing in the central nervous system (CNS), have a profound impact on the maintenance of CNS homeostasis, as they not only function as scavengers to facilitate the removal of invading pathogens or damaged tissues, but also actively regulate neurogenesis, synaptic pruning and neuronal activity (*Colonna and Butovsky, 2017*; *Gomez-Nicola and Perry, 2015*). It is, therefore, not surprising that the dysfunction of microglia is closely linked with the occurrence and progression of many neurodegenerative disorders (*Glass et al., 2010*; *Li et al., 2012*; *Prinz et al., 2011*), thus making it a promising therapeutic target in the prevention and treatment of these diseases.

In mice, fate-mapping methods revealed that, unlike other glia cells with neuroectoderm origin, microglia in adults originate from yolk-sac (YS)-derived primitive macrophages, which colonize the

developing brain rudiment during early embryogenesis (*Ginhoux et al., 2010*; *Schulz et al., 2012*). Whether microglia in adult mammals were contributed by monocyte-derived microglia precursors had long been a debate until the establishment of the parabiosis assay (*Ajami et al., 2007*). It is now well accepted that under physiological conditions, microglia in mice are maintained locally by self-renewal throughout the lifespan. In accordance with this concept, the microglia population was found to be rapidly reconstituted by resident cells after genetic or pharmacological ablation of microglia (*Bruttger et al., 2015*; *Elmore et al., 2014*). However, when estimating the physiological turnover rate of microglia via different methods such as $^3$H-thymidine/BrdU/EdU incorporation, Ki67 staining or two-photon live imaging, different studies yielded distinct or even contradictory results i.e., the estimated microglia turnover rate ranging from 0.13 to 0.8% (*Askew et al., 2017*; *Füger et al., 2017*; *Lawson et al., 1992*; *Tay et al., 2017*), which indicated that microglia in adult mice (~2 years lifespan) were either long-lived without replenishment or replenished for several times during the whole lifespan of the animals. Nonetheless, these studies have reached the consensus that in adult mice, all microglia have the proliferation capability and there are no putative microglia precursors existing to proliferate asymmetrically and continuously to contribute to the microglia pool. Moreover, microglia proliferation had been shown to be spatially and temporally coupled with apoptosis to maintain the homeostasis of their population (*Askew et al., 2017*), which indicates that not only proliferating microglia locate near the dying microglia, but also newborn microglia manifest a higher frequency of apoptosis. However, whether this phenomenon is for the quality control of microglia or has any other physiological relevance remains unclear. In addition, as a fraction of newborn microglia underwent apoptosis and did not contribute to microglia homeostasis, previously calculated proliferation rates might potentially overestimate the true turnover rate of microglia. Collectively, although these studies give some insights into the understanding of microglia turnover and maintenance, our knowledge about the underlying cellular and molecular mechanisms are still limited.

PU.1 (encoded by the *SPI1* gene), the founding member of the spleen focus forming virus proviral integration oncogene (SPI) subfamily of the E-twenty six (ETS) family transcription factor, is one of the best-characterized regulators governing the development of macrophage lineage (*McKercher et al., 1996*; *Scott et al., 1994*). Disruption of PU.1 in mice causes early lethality of the embryos and complete absence of macrophages (*McKercher et al., 1996*; *Scott et al., 1994*), including microglia in the brain (*Kierdorf et al., 2013*). Interestingly, the *SPI1* expression level was found to remain at a high level in microglia after their terminal differentiation (*Walton et al., 2000*), implying that PU.1 also regulates the cellular behaviors and functions of microglia in the sophisticated CNS environment. Indeed, Chromatin immunoprecipitation (ChIP)-Seq analysis with microglia cell line BV2 cells revealed thousands of PU.1 targets (*Satoh et al., 2014*). Accordingly, silencing of PU.1 by siRNA in mixed human glial culture led to the reduced viability of microglia and decreased phagocytosis of amyloid-beta (Aβ42) (*Smith et al., 2013*). However, as a previous study reported that inactivating either 1 copy or 2 copies of *SPI1* by fusing PU.1 with cytoplasm-retaining estrogen receptor (ER) had no effect on the viability of alveolar macrophages (*Karpurapu et al., 2011*), whether PU.1 indeed regulated the survival and maintenance of microglia in the in-vivo conditions remained to be further investigated. Moreover, human SNPs that altered *SPI1* expression level were reported to be associated with Alzheimer's disease (AD) pathogenesis (*Cao et al., 2022*; *Huang et al., 2017*). A recent study showed that conditional inactivation or pharmacological inhibition of PU.1 in the AD mouse model ameliorated neuroinflammation, prevented microgliosis, and improved cognitive performance (*Ralvenius et al., 2023*). Although the reduction of microglia number was considered as a result of the amelioration of neuroinflammation, whether PU.1 directly regulated microglia survival and maintenance to prevent the propagation of neuroinflammation in diseased conditions deserves to be carefully revisited. SPI-B, another member in the mammalian SPI subfamily, has been shown to be dispensable for the development of the macrophage lineage (*Su et al., 1997*). Yet, its involvement in the regulation of microglia turnover and maintenance remains to be investigated.

Owing to the genetic amenability and the feasibility for live imaging and lineage tracing, zebrafish has been adopted as an ideal model for the study of microglia development and function (*Hughes and Appel, 2020*; *Iyer et al., 2022*; *Li et al., 2012*; *Peri and Nüsslein-Volhard, 2008*). Zebrafish microglia are highly conserved with their mammalian counterparts in terms of morphology, molecular signatures, development-regulatory genetic networks, as well as functions (*Xu et al., 2015*; *Yu et al., 2017*). Different from the single origin of mouse microglia, zebrafish microglia were shown

to arise from two distinct sources, i.e., the rostral blood island (RBI), equivalent of mouse YS for primitive myelopoiesis, and the aorta-gonad-mesonephros (AGM) region for hematopoietic stem cell (HSC) initiation. The RBI microglia colonize the developing brain during early embryogenesis, but are gradually replenished by AGM adult microglia (*Xu et al., 2015*). Further study revealed that this dynamic shift of microglia pool is controlled by Il34-Csf1ra signaling-dictated cell competition (*Yu et al., 2023*). Yet, when the microglia pool is established in adult zebrafish, the molecular mechanisms underlying their turnover and maintenance remain incompletely understood. Moreover, although Pu.1 (also called Spi1b in zebrafish, Ensembl:ENSDARG00000000767) and its paralogue Spi-b (also called Spi1a, Ensembl:ENSDARG00000067797), the zebrafish orthologues of mammalian PU.1 and SPI-B, were shown to be indispensable for microglia early development of both origins in zebrafish (*Xu et al., 2015*; *Yu et al., 2017*), their functions in the survival and maintenance of microglia in adult animals have not been established.

In the present study, via the condition knockout strategy to generate a mosaic microglia population in adult zebrafish and mice, we revealed that the turnover and maintenance of microglia is evolutionarily conserved and regulated by Pu.1/Spi1 dosage from teleosts to mammals.

## Results

### Adult microglia in zebrafish undergo rapid turnover and random proliferation to replenish and maintain their pool

Previous studies have shown that there are two major subtypes of myeloid cells, i.e., $ccl34b.1^+$ + and $ccl34b.1^-ccl19a.1^+$ brain-resident dendritic cells (+) in adult zebrafish brain (*Wu et al., 2020*; *Zhou et al., 2023*). To monitor the turnover and maintenance of microglia in adult zebrafish brain, we employed the microglia-specific *TgBAC(ccl34b.1:eGFP)* transgenic reporter line (*Wu et al., 2020*) and performed EdU pulse-chase experiment, a similar strategy with that in the mouse to determine microglia turnover (*Askew et al., 2017*; *Tay et al., 2017*). As a single dose of EdU injection might under-estimate the microglia turnover rate, we intraperitoneally injected EdU into adult *TgBAC(ccl34b.1:eGFP)* fish for either 1, 3, or 5 consecutive days. We then collected the fish brain at 1 day post-injection (dpi) for cryosection and co-staining of eGFP, Lcp1 (a pan leukocyte marker expressed in both microglia and DCs), EdU, and DAPI, in which $ccl34b.1$-$eGFP^+Lcp1^+$ cells are microglia and $ccl34b.1$-$eGFP^-Lcp1^+$ cells represent DCs (*Figure 1A*). Quantification results showed that, while a single dose of EdU pulse labeled 1.892 ± 0.234% microglia, the EdU incorporation rate for microglia after three or five doses of EdU pulses increased almost linearly (*Figure 1B*, green bars). Based on these data, the estimated microglia turnover rate (daily) is ~2.6% (total EdU incorporation rate divided by the times of EdU-pulses) (*Figure 1C*, green bars). On the other hand, the EdU incorporation rate for the DCs was much lower (*Figure 1B–C*, red bars), suggesting that the DCs in the brain are relatively steady with a very low turnover rate.

Previously, via the EdU/BrdU dual-pulse assay, *Tay et al., 2017* have shown that microglia in adult mouse brain proliferate randomly during homeostasis and no putative microglia precursors have been identified accordingly. To clarify whether there might be putative microglial precursors that continuously proliferate to contribute to microglia turnover in zebrafish, we employed a similar strategy, in which adult wild-type (WT) fish were injected with a single dose of EdU followed with five daily BrdU pulses to determine the BrdU proliferation rate of $EdU^+$ and $EdU^-$ microglia (*Figure 1D*, upper panel). Quantification results showed that the BrdU incorporation rates were comparable between $EdU^+$ and $EdU^-$ microglia pools (*Figure 1D–E*). These data strongly suggest that, similar to mammals (*Tay et al., 2017*), microglia in adult zebrafish randomly undergo proliferation and there are no well-defined microglial precursors that continuously proliferate to replenish and maintain the microglia pool.

### Generation and characterization of the visible conditional knockout allele *pu.1^{KI}*

To elucidate the molecular mechanisms underlying microglia turnover and maintenance, we focused on the Ets family transcription factor PU.1/SPI1, which had been shown to be a master regulator for macrophage/microglia development in fish and mammals (*McKercher et al., 1996*; *Scott et al., 1994*; *Yu et al., 2017*). Because macrophages/microglia development is completely blocked at early development in *pu.1*-null zebrafish mutants (*Yu et al., 2017*), we decided to generate a visible *pu.1*

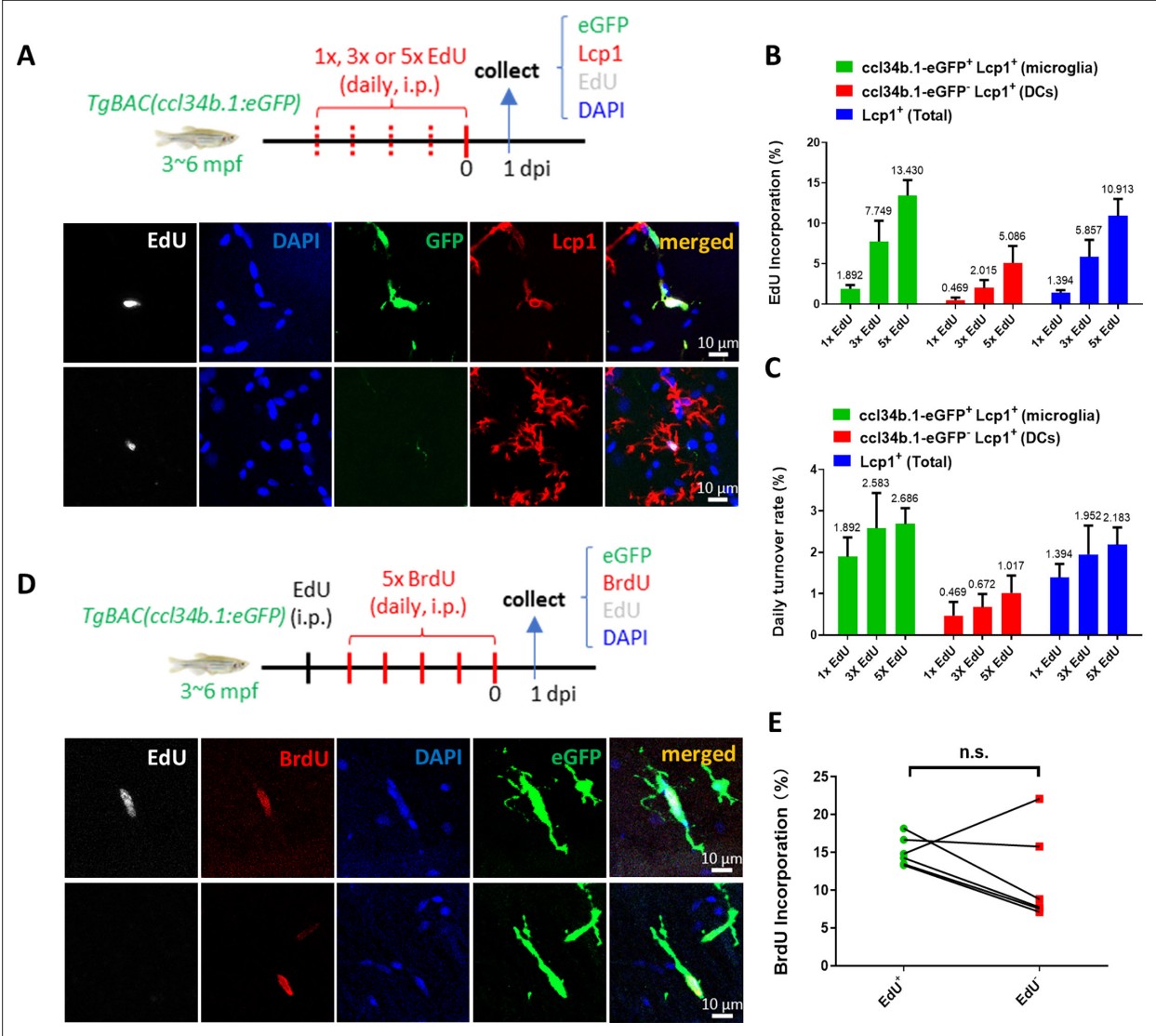

**Figure 1.** Adult microglia in zebrafish undergo rapid turnover and random proliferation to replenish and maintain their pool. (**A**) Schematic diagram shows the workflow of EdU pulse experiment in adult *TgBAC(ccl34b.1:eGFP)* zebrafish and the representative images of proliferating microglia (ccl34b.1-eGFP+ Lcp1+EdU+) and dendritic cells (DCs) (ccl34b.1-eGFP-Lcp1+EdU+) in the midbrain. (**B**) Quantification of the EdU incorporation proportions in microglia and dendritic cells (DCs) with different dosages of EdU pulses. (n=4 for each group) (**C**) The daily turnover rate of microglia and DCs was calculated by dividing the EdU incorporation rate with EdU pulses. (**D**) Schematic diagram shows the experimental setup for EdU-BrdU double-pulse in adult wild-type fish and the representative images show two BrdU+ microglia with or without EdU incorporation. (**E**) Comparison of BrdU incorporation proportions in EdU+eGFP+ and EdU-eGFP+ microglia (n=6). n.s.=not significant, p>0.05.

conditional knockout allele *pu.1KI* to study the role of Pu.1 in microglia maintenance via the Non-homologous end joining (NHEJ)-mediated knock-in method (*Figure 2A*, *Figure 2—figure supplement 1A*; *Li et al., 2015*). As shown in *Figure 2A*, the knock-in DNA fragment consisted of loxP, *pu.1* exon 4–6 followed by the self-cleaving P2A sequence-conjugated eGFP, and a splicing-acceptor site SpA2 followed by the P2A-DsRed. In addition, the 3' downstream sequence of the *pu.1* gene (gray box) was added to the 3' end of each fluorescent protein to ensure efficient termination of RNA transcription and proper expression of the fluorescent proteins.

Three different founders were obtained. PCR and deep sequencing analysis of F1 embryos confirmed the correct integration of donor DNA fragment in the *pu.1* locus (*Figure 2—figure supplement 1B–C*). Since the knock-in *pu.1KI* allele would generate intact Pu.1 and eGFP concurrently (*Figure 2A*), we predicted the expression pattern of eGFP, which presumably represents the expression of the knock-in Pu.1, should completely overlap with the endogenous Pu.1. Indeed, co-staining

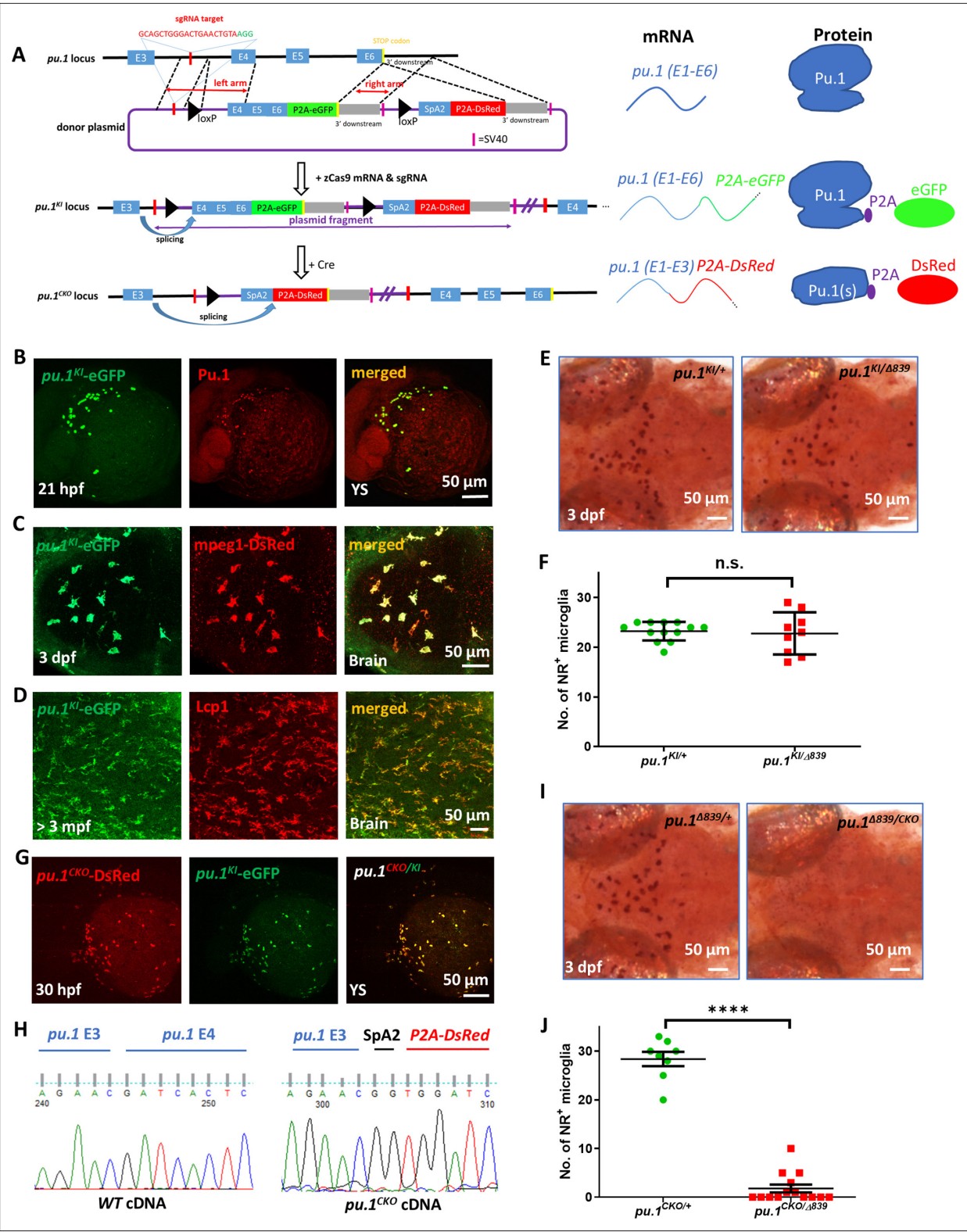

**Figure 2.** Generation and characterization of the visible conditional knockout allele *pu.1^KI^*. (**A**) Schematic diagrams show the generation of *pu.1^KI^* allele and the principle for *pu.1* visible conditional knockout. Briefly, the donor plasmid, which contains: (1) the sgRNA target sequence in *pu.1* intron 3~4; (2) the loxP-flanked *pu.1* coding sequence (exon 4-6) followed by P2A-eGFP; (3) the splicing acceptor site followed by P2A-DsRed sequence was knocked into the endogenous *pu.1* locus via non-homologous end joining (NHEJ) to generate *pu.1^KI^*. In principle, the splicing event occurring between E3 and E4 in *pu.1^KI^* would produce intact Pu.1 and eGFP concurrently. After Cre-mediated recombination, removal of *pu.1* E4-6 and splicing of P2A-DsRed

*Figure 2 continued on next page*

*Figure 2 continued*

sequence in *pu.1^CKO* allele leads to the disruption of Pu.1 and fluorescent color change. (**B**) Co-staining of anti-eGFP and anti-Pu.1 antibodies on the yolk sac (YS) of 21-hpf *pu.1^KI* embryos. (**C**) Fluorescent imaging of the optic tectum (OT) region of 3-dpf *pu.1^KI;Tg(mpeg1:LRLG)* embryos. (**D**) Co-staining of anti-eGFP and anti-Lcp1 antibodies on the midbrain cross section of adult *pu.1^KI* fish. (**E**) Neutral red staining of *pu.1^KI/+* and *pu.1^KI/Δ839* embryos at 3 dpf. (**F**) Quantification of NR⁺ microglia in *pu.1^KI/+* (n=13) and *pu.1 ^KI/Δ839* (n=9) embryos at 3 dpf. (**G**) Fluorescent imaging of the YS region of 30-hpf *pu.1^KI/ CKO* embryos. (**H**) Chromogram of cDNA sequence from wildtype and *pu.1^CKO* embryos shows the precise splicing of *pu.1* E4 and *P2A-DsRed* cassette to *pu.1* E3. (**I**) Neutral red staining of *pu.1^CKO/+* and *pu.1^CKO/Δ839* embryos at 3 dpf. (**J**) Quantification of NR⁺ microglia in *pu.1^CKO/+* (n=8) and *pu.1^CKO/Δ839* embryos at 3 dpf (n=14). n.s.=not significant, p>0.05; ****p<0.0001.

The online version of this article includes the following source data and figure supplement(s) for figure 2:

**Figure supplement 1.** Characterization of *pu.1^KI* and *pu.1^CKO* alleles.

**Figure supplement 1—source data 1.** Original file for blots of *Figure 2—figure supplement 1B*.

**Figure supplement 1—source data 2.** Original blots of *Figure 2—figure supplement 1B* indicating the relevant bands.

of eGFP and Pu.1 antibodies revealed a perfect overlap of the eGFP⁺ cells and Pu.1⁺ cells in *pu.1^KI/+* embryos (*Figure 2B*), suggesting that the *pu.1^KI* allele recapitulates the expression of endogenous *pu.1*. Further studies suggested that *pu.1^KI-eGFP* efficiently labels embryonic microglia, and both microglia and DCs in adult, as revealed by the co-localization of *pu.1^KI-eGFP* and macrophage-specific reporter line *Tg(mpeg1:LRLG)* at 3 dpf, and by the co-staining of eGFP and Lcp1 antibodies on the brain sections of adult *pu.1^KI/+* fish, respectively (*Figure 2C and D*). To further confirm functional intactness of the *pu.1^KI* allele, we crossed the *pu.1^KI* fish with the *pu.1^Δ839* mutants to examine if the *pu.1^KI* allele is capable of rescuing the microglia phenotype. Indeed, quantification of microglia number revealed no difference between the *pu.1^Δ839/KI* and *pu.1^Δ839/+* embryos (*Figure 2E and F*), indicating that the *pu.1^KI* allele functionally recapitulates the endogenous *pu.1*.

To test whether the Cre-mediated excision occurs correctly in the *pu.1^KI* allele, we injected Cre mRNA into the *pu.1^KI* embryos at one-cell stage. The injected embryos were raised to adulthood (*pu.1^CKO* fish) and crossed with the *pu.1^KI* fish to generate *pu.1^CKO/KI* fish. As illustrated in *Figure 2A*, the Cre-mediated excision should remove the E4-6-P2A-eGFP cassette and allow the proper splicing of the splicing-acceptor site SpA2 and joining of the P2A-DsRed with exon 3 (*Figure 2A*), leading to the induction of DsRed expression. Indeed, fluorescent imaging of the *pu.1^CKO/KI* embryos revealed robust DsRed (*pu.1^CKO* allele) signals perfectly overlapping with the eGFP (*pu.1^KI* allele) signals (*Figure 2G*), suggesting the correct DNA excision and RNA splicing in the *pu.1^CKO* allele. This conclusion was further validated by sequencing analysis of the *pu.1^CKO* transcripts (*Figure 2H*). Since the endogenous *pu.1* exons were not replaced, but instead they were separated far away from the upstream exons by donor plasmid, we performed quantitative RT-PCR analysis of *pu.1^CKO/CKO* embryos to determine the possible expression of the endogenous *pu.1* exons in *pu.1^CKO* allele. The result showed that the relative expression of endogenous *pu.1* exons is around 1–2% in comparison with that of DsRed expression (*Figure 2—figure supplement 1D*), suggesting that expression of *pu.1* is largely abolished in *pu.1^CKO* allele. To phenotypically characterize *pu.1^CKO* allele, and to avoid incomplete inactivation of Pu.1 function in downstream microglia tracing experiments, which may occur when using two *pu.1^KI* alleles due to the lower CreER recombination efficiency in zebrafish compared to mice, we outcrossed the *pu.1^CKO* fish with the *pu.1^Δ839* null mutants. As anticipated, the development of embryonic microglia was completely blocked in the *pu.1^CKO/Δ839* embryos (*Figure 2I and J*), a phenotype recapitulating the defect of the *pu.1^Δ839* homozygous mutants (*Yu et al., 2017*). These data demonstrate that the *pu.1^KI* allele works very well and serves as a useful tool to explore the role of Pu.1 in adult microglia survival and maintenance.

## Pu.1 and Spi-b regulate the turnover and maintenance of microglia in a dosage-dependent manner

To investigate the role of Pu.1 in the survival and maintenance of microglia in adult zebrafish, we crossed the *pu.1^KI* fish with the leukocyte-specific *Tg(coro1a:CreER)* line and the *pu.1^Δ839* mutants to generate *pu.1^KI/+;Tg(coro1a:CreER)* and *pu.1^KI/Δ839;Tg(coro1a:CreER)* fish. We then intraperitoneally injected three doses of 4-OHT into these adult fish and collected the brain samples at 2 dpi and 8 dpi for the quantification of microglia number after cryo-section and eGFP/DsRed immuno-staining (*Figure 3A*, upper panel). As shown in *Figure 3A* (lower panel), DsRed⁺ cells were readily detected in

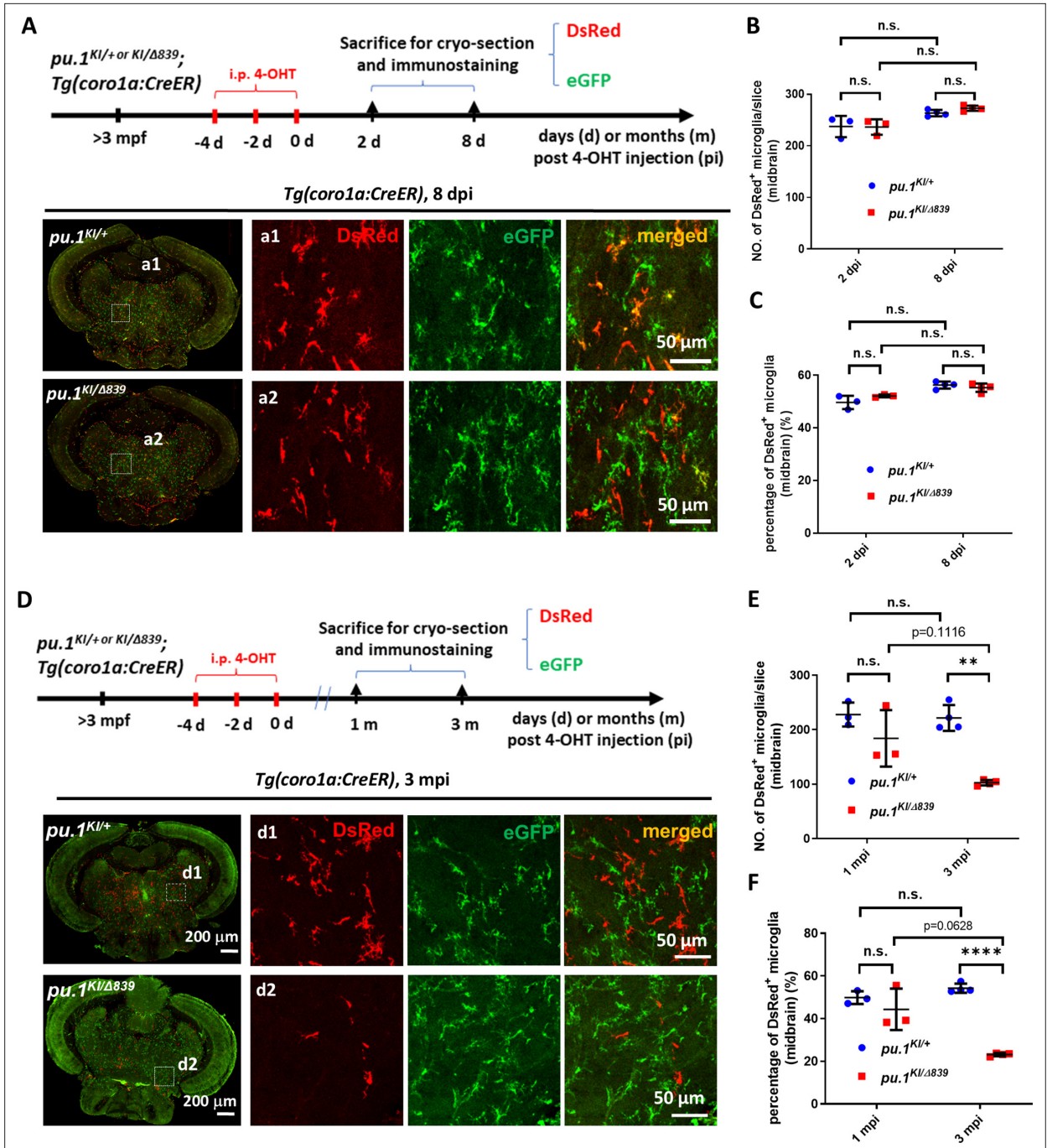

**Figure 3.** *pu.1*-deficient microglia were chronically eliminated in mosaic condition. (**A**) The experimental setup for *pu.1* conditional knockout in adult zebrafish and the representative images of midbrain cross section of *pu.1^KI/+^;Tg(coro1a:CreER)* and *pu.1^KI/Δ839^;Tg(coro1a:CreER)* fish at 8 days post 4-OHT injection (dpi). (**B**) Quantification of the number of DsRed⁺ microglia on the midbrain cross section of *pu.1^KI/+^;Tg(coro1a:CreER)* and *pu.1^KI/Δ839^;Tg(coro1a:CreER)* fish at 2 dpi (n=3) and 8 dpi (n=4). (**C**) Quantification of the proportion of DsRed⁺ microglia on the midbrain cross section of *pu.1^KI/+^;Tg(coro1a:CreER)* and *pu.1^KI/Δ839^;Tg(coro1a:CreER)* fish at 2 dpi (n=3) and 8 dpi (n=4). (**D**) The experimental setup for *pu.1* conditional knockout in adult zebrafish and the representative images of midbrain cross section of *pu.1^KI/+^;Tg(coro1a:CreER)* and *pu.1^KI/Δ839^;Tg(coro1a:CreER)* fish at 3 months post 4-OHT injection (mpi). (**E**) Quantification of the number of DsRed⁺ microglia on the midbrain cross section of *pu.1^KI/+^;Tg(coro1a:CreER)* and *pu.1^KI/Δ839^;Tg(coro1a:CreER)* fish at 1 mpi (n=3) and 3 mpi (n=4). (**F**) Quantification of the proportion of DsRed⁺ microglia on the midbrain cross section of *pu.1^KI/+^;Tg(coro1a:CreER)* and *pu.1^KI/Δ839^;Tg(coro1a:CreER)* fish at 1 mpi (n=3) and 3 mpi (n=4). n.s.=not significant, p>0.05; **p<0.01; ****p<0.0001.

The online version of this article includes the following figure supplement(s) for figure 3:

**Figure supplement 1.** Conditional depletion of Pu.1 in embryonic microglia had no effect on their short-term survival.

*Figure 3 continued on next page*

*Figure 3 continued*

**Figure supplement 2.** Microglia number is not affected in *pu.1^Δ839^* null mutants.

**Figure supplement 3.** Simultaneous inactivation of Pu.1 and Spi-b leads to rapid elimination of microglia in zebrafish.

**Figure supplement 4.** Pu.1/Spi-b-deficient microglia undergo apoptosis in zebrafish.

**Figure supplement 5.** Conditional inactivation of Pu.1 leads to chronic elimination of microglia in the spinal cord and retina of adult zebrafish.

**Figure supplement 6.** Conditional inactivation of Pu.1 leads to chronic elimination of microglia in the brain of adult zebrafish.

the brains of both *pu.1^KI/+^;Tg(coro1a:CreER)* and *pu.1^KI/Δ839^;Tg(coro1a:CreER)* fish at 8 dpi. Quantification of the number and proportion of DsRed⁺ cells at 2 dpi and 8 dpi revealed no difference between *pu.1^KI/Δ839^;Tg(coro1a:CreER)* mutant fish and *pu.1^KI/+^;Tg(coro1a:CreER)* control siblings (*Figure 3B–C*). Similarly, conditional knockout of *pu.1* in embryonic microglia also showed no effect on microglial survival (*Figure 3—figure supplement 1*), which contrasts to the elimination of microglia during early development by global knockout of *pu.1* due to Pu.1's essential role in myeloid lineage specification. In addition, we also found that the number and proportion of microglia (ccl34b.1-eGFP⁺ Lcp1⁺) and DCs (ccl34b.1-eGFP⁻Lcp1⁺) were comparable between *pu.1^Δ839/Δ839^;TgBAC(ccl34b.1:eGFP)* null mutants and *pu.1^Δ839/+^;TgBAC(ccl34b.1:eGFP)* siblings (*Figure 3—figure supplement 2*). These data suggest that Pu.1 is not required for the survival of microglia and DCs. We reasoned that the survival of *pu.1*-deficient microglia is likely attributed to the presence of *spi-b* (the paralogue of *pu.1* in zebrafish, also called *spi1a*), which has been shown to play a compensatory role for Pu.1 in microglia early development (*Yu et al., 2017*). Indeed, conditional inactivation of Pu.1 in the *spi-b^Δ232/Δ232^* null mutants led to a rapid depletion of microglia by apoptosis within several days (*Figure 3—figure supplements 3 and 4*), indicating that Pu.1 and Spi-b act redundantly to regulate the survival of microglia.

To further investigate whether Pu.1 deficiency might have a long-term effect on the turnover and maintenance of microglia, we traced *pu.1*-deficient DsRed⁺ cells to 1 month post-injection (mpi), and 3 mpi after conditionally inactivating Pu.1 in adult *pu.1^KI/+^;Tg(coro1a:CreER)* and *pu.1^KI/Δ839^;Tg(coro1a:CreER)* fish (*Figure 3D*, upper panel). Intriguingly, we found that the number and proportion of DsRed⁺ cells in the midbrains of the *pu.1^KI/Δ839^;Tg(coro1a:CreER)* fish began to decline at 1 mpi and were further reduced, becoming significantly lower than that in *pu.1^KI/+^;Tg(coro1a:CreER)* fish by 3 mpi (*Figure 3E–F*). Likewise, the number and proportion of DsRed⁺ cells in the retina and spinal cord were also significantly reduced in the *pu.1^KI/Δ839^;Tg(coro1a:CreER)* fish by 3 mpi (*Figure 3—figure supplement 5*). Since microglia are the major population of the *coro1a⁺* cells in the brain and spinal cord and the exclusive population in the retina (*Wu et al., 2020*), the reduction of DsRed⁺ cells in the *pu.1^KI/Δ839^;Tg(coro1a:CreER)* fish at 1 mpi and 3 mpi are mainly attributed to the loss of microglia. Based on the amoeboid and ramified morphology, microglia and DCs could be distinguished with about 90% accuracy (*Figure 3—figure supplement 6A–B*). Thus, to further confirm above conclusion, we quantified microglia and DCs separately in 3-mpi *pu.1^KI/+^;Tg(coro1a:CreER)* and *pu.1^KI/Δ839^;Tg(coro1a:CreER)* fish. As anticipated, *pu.1*-deficient microglia were significantly reduced at 3 mpi (*Figure 3—figure supplement 6C–D*). These results suggest that, although *pu.1* is dispensable for the immediate survival of microglia (*Figure 3B–C*), it is essential for the turnover and long-term maintenance of microglia in adult zebrafish, which could not be fully compensated by *spi-b*.

## *pu.1*-deficient microglia in mosaic condition were eliminated by cell competition

Given the fact that *pu.1*-deficient microglia are gradually eliminated in mosaic condition (*Figure 3E–F*), but their number remains relatively normal in *pu.1^Δ839^* null mutants (*Yu et al., 2017*), we reasoned that the elimination of the *pu.1*-deficient microglia in mosaic condition is likely attributed to the cell competition between *pu.1*-deficient DsRed⁺ microglia and their neighboring eGFP⁺ sibling cells harboring WT *pu.1*. To test this hypothesis, we compared the cell death and proliferation rates of microglia by TUNEL and BrdU incorporation assays in both conditions, respectively. To determine the cell death and proliferation rates of *pu.1*-deficient microglia in mosaic condition, we treated the adult *pu.1^KI/Δ839^;Tg(coro1a:CreER)* fish with 4-OHT to inactivate Pu.1. At 26 dpi, the 4-OHT-treated fish were then injected with five doses (one dose daily) of BrdU and the brain samples were collected 1 day later for TUNEL and BrdU staining, respectively (*Figure 4A*). Quantification results showed that the percentage of TUNEL⁺DsRed⁺ microglia (*pu.1*-deficient cells) in the brains of the *pu.1^KI/Δ839^;Tg(coro1a:CreER)* fish

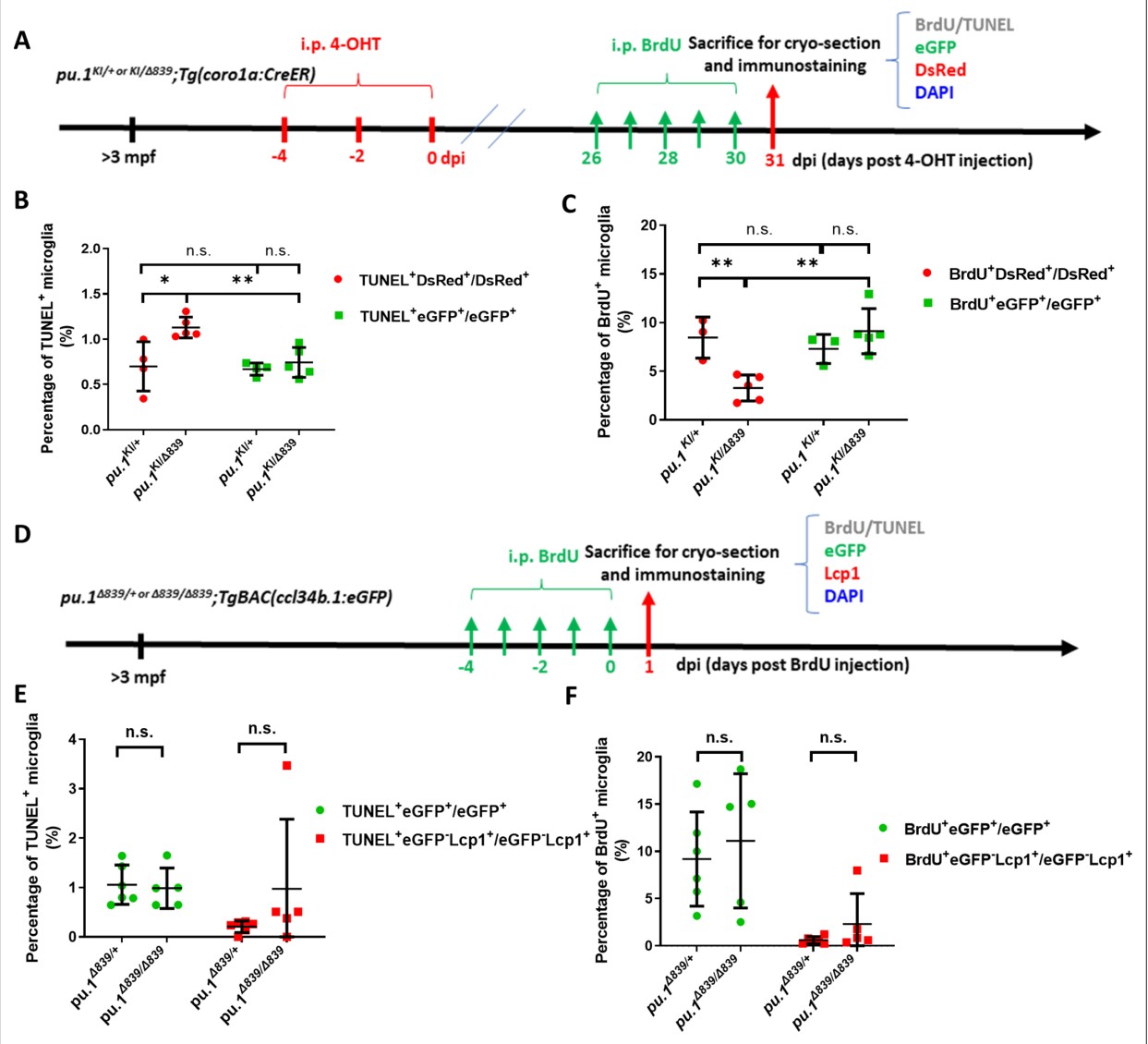

**Figure 4.** *pu.1*-deficient microglia in mosaic condition were eliminated by cell competition. (**A**) The experimental setup for BrdU incorporation and TUNEL assays in adult *pu.1^{KI/+}*;*Tg(coro1a:CreER)* and *pu.1^{KI/Δ839}*;*Tg(coro1a:CreER)* fish at 1 mpi. (**B**) Quantification of the percentage of TUNEL^+ cells in eGFP^+ and DsRed^+ microglia in *pu.1^{KI/+}*;*Tg(coro1a:CreER)* (n=4) and *pu.1^{KI/Δ839}*;*Tg(coro1a:CreER)* (n=5) fish at 1 mpi. (**C**) Quantification of the percentage of BrdU^+ cells in eGFP^+ and DsRed^+ microglia in adult *pu.1^{KI/+}*;*Tg(coro1a:CreER)* (n=4) and *pu.1^{KI/Δ839}*;*Tg(coro1a:CreER)* fish at 1 mpi. (**D**) The experimental setup for BrdU incorporation and TUNEL assays in adult *pu.1^{Δ839/+}*;*TgBAC(ccl34b.1:eGFP)* and *pu.1^{Δ839/Δ839}*;*TgBAC(ccl34b.1:eGFP)* fish. (**E**) Quantification of the percentage of TUNEL^+eGFP^+ microglia in adult *pu.1^{Δ839/+}*;*TgBAC(ccl34b.1:eGFP)* and *pu.1^{Δ839/Δ839}*;*TgBAC(ccl34b.1:eGFP)* fish. (**F**) Quantification of the percentage of BrdU^+eGFP^+ microglia in adult *pu.1^{Δ839/+}*;*TgBAC(ccl34b.1:eGFP)* (n=6) and *pu.1^{Δ839/Δ839}*;*TgBAC(ccl34b.1:eGFP)* (n=5) fish. n.s.=not significant, p>0.05; *p<0.05; **p<0.01.

was significantly higher than that of TUNEL^+eGFP^+ cells (*Figure 4B*). In this assay, the 4-OHT-treated *pu.1^{KI/+}*;*Tg(coro1a:CreER)* fish were included as the control, and as expected, the percentages of TUNEL^+ cells in the DsRed^+ and eGFP^+ microglia pools in control fish were comparable (*Figure 4B*). In parallel, we also found that the BrdU incorporation rates of *pu.1*-deficient microglia (DsRed^+) in the 4-OHT-treated *pu.1^{KI/Δ839}*;*Tg(coro1a:CreER)* fish was significantly lower than that of eGFP^+ cells (*Figure 4C*), while the BrdU incorporation rates of DsRed^+ cells and eGFP^+ cells in the control *pu.1^{KI/+}*;*Tg(coro1a:CreER)* fish was comparable (*Figure 4C*). These results indicate that conditional inactivation of Pu.1 in adult zebrafish leads to the impairment of cell proliferation and accelerated cell death of microglia in mosaic condition. To estimate the cell death and proliferation rates of microglia in *pu.1*-null mutants, we utilized a similar strategy via pulsing the adult *pu.1^{Δ839/+}*;*TgBAC(ccl34b.1:eGFP)* or

pu.1$^{\Delta839/\Delta839}$;TgBAC(ccl34b.1:eGFP) fish with five doses of BrdU (daily) and collecting the brain samples at 1 day later for TUNEL and BrdU staining, respectively (**Figure 4D**). As shown in **Figure 4E**, the percentage of TUNEL$^+$ccl34b.1-eGFP$^+$ microglia showed no difference between pu.1$^{\Delta839/\Delta839}$ mutants and pu.1$^{\Delta839/+}$ siblings, demonstrating that the intrinsic viability of pu.1-deficient microglia in the null mutants was largely unaffected. Similarly, the proliferation capability (as indicated by BrdU incorporation rate) of pu.1-deficient microglia in the null mutants remained normal as well in comparison with control siblings (**Figure 4F**). Collectively, these results strongly suggest that the chronic elimination of pu.1-deficient DsRed$^+$ microglia is indeed caused by cell competition, which only occurs in the mosaic condition when the eGFP$^+$ microglia harboring a single copy of pu.1 are present.

## The out-competition of *pu.1*-deficient microglia in mosaic condition is mediated by Tp53

To uncover the key downstream targets that mediate the out-competition of pu.1-deficient DsRed$^+$ microglia in mosaic condition, we manually picked DsRed$^+$ and eGFP$^+$ microglia from the 4-OHT-treated pu.1$^{KI/\Delta839}$;Tg(coro1a:CreER) fish at 3 mpi and performed transcriptomic analysis (**Figure 5A**). To minimize the contamination of DCs, we manually picked 3–5 DsRed$^+$ or eGFP$^+$ cells from the brain suspension of 3-mpi pu.1$^{KI/\Delta839}$;Tg(coro1a:CreER) fish in each tube for cDNA library construction and RNAseq. Samples with strong expression of microglia-related genes and faint DC-related genes were chosen for further analysis (**Figure 5—figure supplement 1A**). Among the differentially expressed genes (DEGs), the tumor suppressor gene tp53 was one of the top-targeted genes robustly upregulated in pu.1-deficient DsRed$^+$ microglia (**Figure 5B-C**, **Figure 5—figure supplement 1B**). Interestingly, recent studies have revealed that, in addition to regulating cell cycle arrest and apoptosis in DNA damage response (**Ou and Schumacher, 2018**; **Vaddavalli and Schumacher, 2022**), Tp53 is also involved in cell competition regulation in both hematopoietic and non-hemopoietic tissues (**Bondar and Medzhitov, 2010**; **Zhang et al., 2017**). Hence, we speculated that the cell competition-mediated chronic elimination of pu.1-deficient microglia was likely dependent on Tp53. To support this hypothesis, we conditionally inactivated Pu.1 to generate mosaic microglia populations in the tp53-deficient mutant background and asked if loss of Tp53 could rescue the microglia phenotype (**Figure 5D**, upper panel). Indeed, results showed that the number of pu.1-deficient DsRed$^+$ microglia in the 4-OHT-treated pu.1$^{KI/\Delta839}$;tp53$^{-/-}$;Tg(coro1a:CreER) fish was significantly restored compared with pu.1$^{KI/\Delta839}$;Tg(coro1a:CreER) control fish (**Figure 5D–F**). Collectively, these results demonstrate that the chronic elimination of pu.1-deficient microglia in mosaic condition is mediated, at least in part, by the elevation of Tp53 activity, which leads to the reduced proliferation and excessive death of these microglia.

## Dosage-dependent regulation of microglia maintenance by PU.1 is evolutionarily conserved from zebrafish to mice

The above data have demonstrated that the long-term maintenance of microglia in adult zebrafish is regulated by Pu.1 in a dosage-dependent manner. We next wondered whether this mechanism is evolutionarily conserved from teleosts to mammals. To address this issue, we first examined the expression of Spi1/Pu.1 (Ensembl: ENSMUSG00000002111) and Spi-b (Ensembl: ENSMUSG00000008193) in adult murine microglia. Quantitative RT-PCR revealed that in contrast to the robust expression of Spi1, Spi-b was barely detectable in microglia (**Figure 6—figure supplement 1A**). It appears that Spi-b has acquired lineage-specific roles during evolution and becomes absent in murine microglia. We, therefore, only focused on Spi1 and generated a Spi1$^{Fl}$ allele, in which the exon 2 of Spi1 gene is flanked by loxP (**Figure 6—figure supplement 1B–C**). The Spi1$^{Fl}$ mice were then crossed with the Cx3cr1$^{CreER-YFP}$ strain (referred to as Cx3cr1$^{CreER}$ hereafter) (**Yona et al., 2013**) to generate Spi1$^{Fl/+}$;Cx3cr1$^{CreER}$ and Spi1$^{Fl/Fl}$;Cx3cr1$^{CreER}$ mice for conditional PU.1 inactivation study (**Figure 6A**).

To investigate whether microglia maintenance in adult mice requires PU.1, we intraperitoneally injected tamoxifen (TAM) (**Askew et al., 2017**) into the Spi1$^{Fl/+}$;Cx3cr1$^{CreER}$ and Spi1$^{Fl/Fl}$;Cx3cr1$^{CreER}$ mice, and collected the brain samples at 7 dpi for cryo-section and IBA1 staining (**Figure 6A**, upper panel). As expected, without TAM, the number of the IBA1$^+$ microglia showed no difference in the brain of Spi1$^{Fl/+}$;Cx3cr1$^{CreER}$ and Spi1$^{Fl/Fl}$;Cx3cr1$^{CreER}$ mice (**Figure 6—figure supplement 1D–E**). However, after TAM injection, the number of the IBA1$^+$ microglia in Spi1$^{Fl/Fl}$;Cx3cr1$^{CreER}$ mice were drastically reduced compared to that in Spi1$^{Fl/+}$;Cx3cr1$^{CreER}$ mice (**Figure 6A–C**), suggesting that PU.1

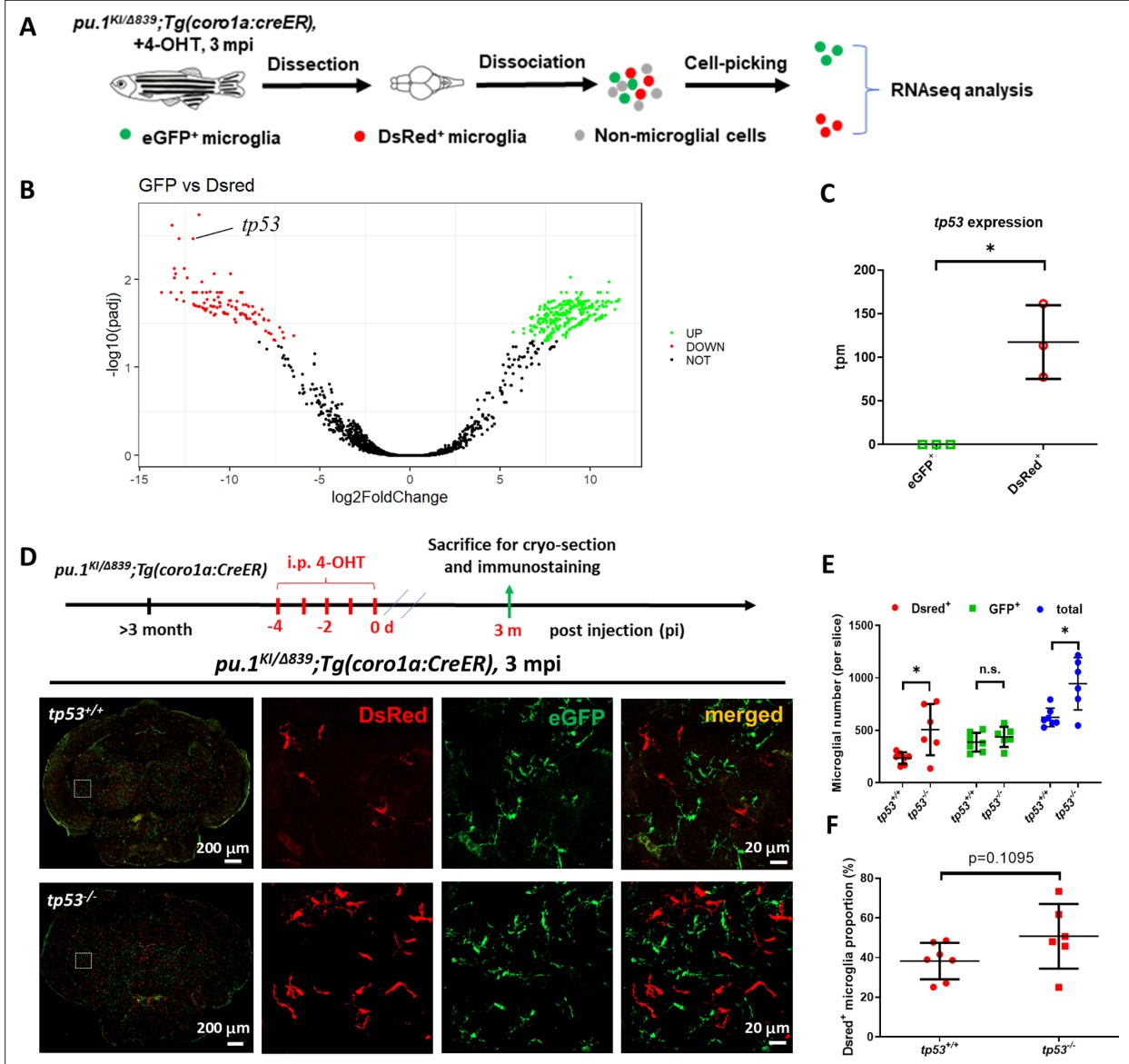

**Figure 5.** Inactivation of Tp53 largely restored the number of *pu.1*-deficient microglia in mosaic condition. (**A**) The experimental setup for the isolation of eGFP⁺ and DsRed⁺ microglia from *pu.1^KI/Δ839^;Tg(coro1a:CreER)* adult brain at 3 mpi for transcriptomic analysis. (**B**) The volcano plot of differentially expressed genes (DEG) between eGFP⁺ and DsRed⁺ microglia at 3 mpi. (**C**) Relative expression of *tp53* in eGFP⁺ (n=3) and DsRed⁺ (n=3) microglia at 3 mpi by transcripts per million (TPM). (**D**) The experimental setup for *pu.1* conditional knockout in wild-type and *tp53^-/-^* adult zebrafish, and the representative images of midbrain cross section of *pu.1^KI/Δ839^;Tg(coro1a:CreER)* and *pu.1^KI/Δ839^;tp53^-/-^;Tg(coro1a:CreER)* fish at 3 mpi. (**E**) Quantification of the number of DsRed⁺, eGFP⁺, and total (DsRed + eGFP) microglia in *pu.1^KI/Δ839^;Tg(coro1a:CreER)* (n=7) and *pu.1^KI/Δ839^;tp53^-/-^;Tg(coro1a:CreER)* (n=6) fish at 3 mpi. (**F**) Quantification of the proportion of DsRed⁺ microglia in *pu.1^KI/Δ839^;Tg(coro1a:CreER)* (n=7) *and pu.1^KI/Δ839^;tp53^-/-^;Tg(coro1a:CreER)* (n=6) fish at 90 dpi. n.s.=not significant, p>0.05; *p<0.05.

The online version of this article includes the following figure supplement(s) for figure 5:

**Figure supplement 1.** RNA-seq analysis of *pu.1*-deficient microglia.

**Figure supplement 2.** In-silico analysis of Pu.1 binding sites on the promoter region of *tp53*.

**Figure supplement 3.** *csf1ra* expression does not decrease after conditional inactivation of Pu.1.

is indeed required for the survival of adult microglia, which is similar to the double depletion of *pu.1* and *spi-b* in zebrafish (***Figure 3—figure supplement 4***). To further explore whether partial loss of PU.1 activity could reduce the competitiveness of microglia and impairs the long-term maintenance of microglia in mice, we intraperitoneally injected sub-dosage of TAM into adult *Spi1^Fl/+^;Cx3cr1^CreER^*

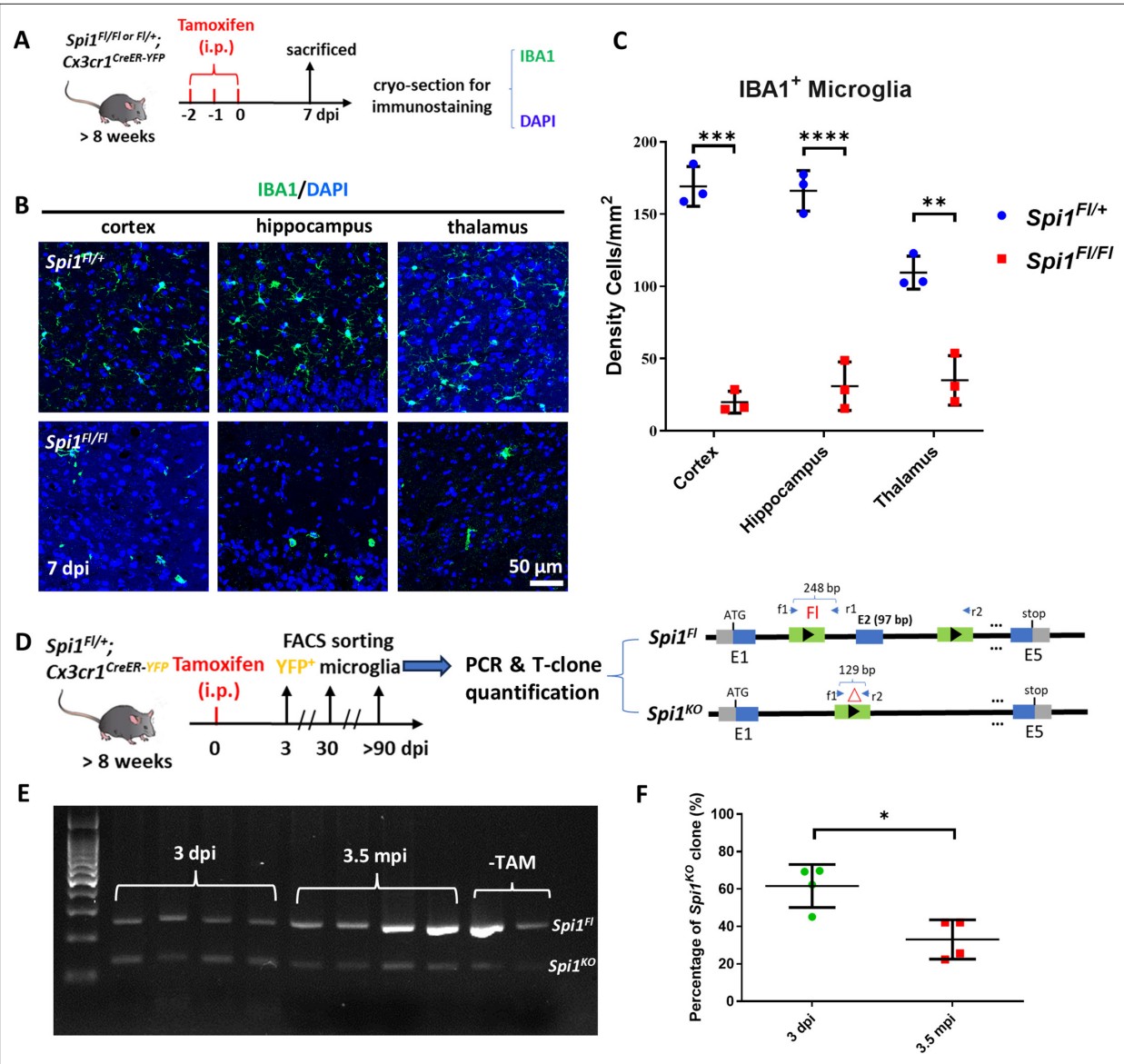

**Figure 6.** Dosage-dependent regulation of microglia maintenance by Pu.1/Spi1 is evolutionary conserved in mice. (**A**) The experimental setup for *Spi1* conditional knockout in adult mice. (**B**) Representative images of IBA1 and DAPI co-staining in the cortex, hippocampus, and thalamus of *Spi1*[Fl/+];*Cx3cr1*[CreER] and *Spi1*[Fl/Fl];*Cx3cr1*[CreER] mice at 7 dpi. (**C**) Quantification of the density of IBA1[+] microglia in the cortex, hippocampus, and thalamus of *Spi1*[Fl/+];*Cx3cr1*[CreER] (n=3) and *Spi1*[Fl/Fl];*Cx3cr1*[CreER] (n=3) mice at 7 dpi. (**D**) The experimental setup for conditional knockout of *Spi1* in *Spi1*[Fl/+];*Cx3cr1*[CreER] mice, and the subsequent PCR detection and T-clone quantification of *Spi1*[Fl] and *Spi1*[KO] alleles in sorted YFP[+] microglia. (**E**) Gel image shows the relative intensity of amplified DNA bands of *Spi1*[Fl] and *Spi1*[KO] alleles in microglia sorted from *Spi1*[Fl/+];*Cx3cr1*[CreER] mice at different stages post-tamoxifen (TAM) injection. (**F**) Quantification of the percentage of *Spi1*[KO] allele at 3 dpi (n=4) and 3.5 mpi (n=4) by T-clone assay. *p<0.05; **p<0.01; ***p<0.001; ****p<0.0001.

The online version of this article includes the following source data and figure supplement(s) for figure 6:

**Source data 1.** Original file for blots of *Figure 6E*.

**Source data 2.** Original blots of *Figure 6E* indicating the relevant bands and treatments.

**Figure supplement 1.** Adult microglia are not affected in *Spi1*[Fl/Fl];*Cx3cr1*[CreER] mice.

**Figure supplement 1—source data 1.** Original file for blots of *Figure 6—figure supplement 1C*.

**Figure supplement 1—source data 2.** Original blots of *Figure 6—figure supplement 1C* indicating the relevant bands.

mice to generate mosaic microglia populations for long-term tracing. To quantify the proportion of $Spi1^{KO/+}$ microglia and $Spi1^{Fl/+}$ microglia, we sorted YFP⁺ microglia from TAM-injected $Spi1^{Fl/+};Cx\text{-}3cr1^{CreER}$ mice at 3 dpi and 3.5 mpi, the timeline similar to previous research in zebrafish. Then we amplified $Spi1^{KO}$ and $Spi1^{Fl}$ alleles in the gDNA pool by PCR for T-clone quantification (*Figure 6D*). As shown in *Figure 6E–F*, while the percentage of $Spi1^{KO/+}$ microglia was comparable to $Spi1^{Fl/+}$ microglia at 3 dpi, it was dramatically reduced by 3.5 mpi, suggesting that removal of a single copy of *Spi1* leads to the chronic elimination of $Spi1^{KO/+}$ microglia, which appears to occur only in the presence of the WT $Spi1^{Fl/+}$ microglia, as *Spi1* heterozygous mice show no difference in the viability of tissue resident macrophages (*Karpurapu et al., 2011*). Collectively, these results indicate that the dosage-dependent cell competition-mediated regulation of microglia maintenance by SPI1/PU.1 is evolutionarily conserved from teleosts to mammals.

## Discussion

In the present study, to explore the role of Pu.1 in microglia survival and maintenance, we established a visible conditional knockout $pu.1^{KI}$ allele in zebrafish via the non-homologous end joining (NHEJ)-mediated knock-in method. Although conditional inactivation of targeted genes has been reported by several independent studies (*Li et al., 2015*; *Li et al., 2019*; *Shin et al., 2023*), it has not been widely applied in zebrafish, possibly due to the difficulty or low efficiency to generate loxP-flanked alleles. Compared to homologous recombination (HR), NHEJ-mediated knock-in of reporter gene has been shown to display 10-fold higher efficiency (*Auer et al., 2014*; *Li et al., 2015*). In our case, the germline transmission to F1 generation for $pu.1^{KI}$ allele is around 15% (*Figure 2—figure supplement 1*). Meanwhile, in our investigation, we noted that the efficiency of Cre-mediated excision of loxP sites at the $pu.1$ locus in zebrafish (*Figure 3*) appears much lower than that in mouse (*Figure 6*). Although it remains to be further validated whether this observation is a general phenomenon, it does highlight the necessity and significance of chimera study of interested genes. In our $pu.1^{KI}$ allele, the expression of DsRed directly indicates Cre-mediated excision and helps us distinguish the mutant cells from their sibling cells, which serves as a good example for chimera study in the future. Moreover, our knock-in allele can also be simply regarded as a loxP-flanked fluorescent reporter line, which not only facilitates the labeling of cells that express the gene of interest, but also can be used in combination with the LEGO-IR system (*Xu et al., 2015*) to trace the fate of these cells. Thus, due to the high integration efficiency of donor DNA fragment and the multifaceted applications of the labeling strategy, we anticipated that our knock-in system would be a powerful tool and broadly adopted in the fields of genetics, developmental biology and cell biology for the chimeric study of interested genes and fate mapping of interested cells in the future.

Via EdU pulse-chase labeling assay, we found that the daily turnover rate of microglia in adult zebrafish is about 2.5%, much higher than that reported in mice and human (*Lopez-Atalaya et al., 2018*). These disparities might be explained by several reasons. First, in our calculation of microglia turnover rate, we did not factor in the unknown cell cycle length of zebrafish microglia. It is possible that the duration of microglia proliferation in zebrafish is much shorter than that in mice. If microglia proliferation takes less than 24 hr in zebrafish, then the calculated daily turnover rate of 2.5% may be overestimated. Second, it is well recognized that zebrafish manifest much stronger regeneration capability than mammals. As such, differentiated cells, like cardiomyocytes and hair cells in zebrafish are in general more proliferative than those in mammals (*Chen et al., 2019*; *Marques et al., 2019*). In this regard, microglia in zebrafish might be pre-coded both genetically and epigenetically to be more proliferative. Finally, microglia turnover could potentially be affected by the distinct efferocytotic burdens via the lysosome pathway, as lysosome has been known to be a convergent point for many biological process, including metabolism, cellular senescence, DNA repair, ER stress and even apoptosis (*Franco-Juárez et al., 2022*; *Perera and Zoncu, 2016*). Indeed, efferocytosis of apoptotic cells has previously been shown to elicit NOX2 assembly and promote ROS production (*Yvan-Charvet et al., 2010*), which in turn triggers cell death via induction of lysosome damage (*Patra et al., 2023*; *Wang et al., 2018*). The more regenerative environment in zebrafish might be accompanied by higher frequency of cell death, which could lead to an increase in efferocytotic burdens of microglia, thereby accelerating the turnover of microglia. Further studies will be required to clarify this issue.

Our results revealed that the survival and maintenance of microglia are highly sensitive to the dosage of PU.1/SPI1 in both teleosts and mammals. Interestingly, a previous study reported that

hetero- or homozygosity of PU/ER(T) allele, in which the ER ligand binding domain-G525R is fused with the full-length PU.1 to block its nuclear translocation, showed no effect on the viability of alveolar macrophages (*Karpurapu et al., 2011*). This contradictory result might potentially result from the leakiness of ER, which was incapable to fully retain PU.1 in cytoplasm. A recent work found that conditional inactivation of PU.1 by feeding the animals with tamoxifen-containing diet in the Alzheimer's disease model CK-p25 mice led to a 33% reduction in microglia number at their young ages. Although the authors considered prevention of microgliosis as a result of the amelioration of neuroinflammation (*Ralvenius et al., 2023*), our data implies that reduction in microglia could also been interpreted by the direct involvement of PU.1 in microglia survival and maintenance, as our data clearly demonstrate that conditional ablation of the PU.1 in mice or Pu.1/Spi-b in zebrafish leads to the rapid depletion of microglia in the brain (*Figure 6*; *Figure 3—figure supplement 4*). As PU.1 is a master regulator that controls the expression of series of genes required for macrophage/microglia development (*Satoh et al., 2014*), rapid death of *Spi1*-null microglia might probably be caused by the orchestration of multiplex factors, including the downregulation of colony-stimulating factor 1 receptor (*Csf1r*) (the essential regulator controlling the survival and maintenance of microglia in both mice and zebrafish) (*Elmore et al., 2014*; *Yu et al., 2023*), the loss of anti-apoptotic genes, such as BCL2A1 (*Jenal et al., 2010*), and absence of other genes that maintain the integrity of cells. Intriguingly, differing from the rapid death of microglia after complete inactivation of PU.1 or Pu.1/Spi-b, microglia with a partial loss of PU.1 or Pu.1 are chronically eliminated only in the mosaic condition via cell competition, in which the microglia with WT *Spi1* or a single copy of *pu.1* are present (*Figures 3 and 6*). Moreover, we further showed that the chronic elimination of *pu.1*-deficient microglia in zebrafish is mediated by Tp53-dependent cell competition, which has been shown to depend on the relative level of Tp53 in competing cells (*Bondar and Medzhitov, 2010*; *Zhang et al., 2017*). A previous study by Tschan et al has shown that PU.1 attenuates the transcriptional activity of the *p53* tumor suppressor family through direct binding to the DNA-binding and/or the oligomerization domains of p53/p73 proteins (*Tschan et al., 2008*). Via in-silico analysis of *tp53* promoter region in zebrafish, we also found three PU.1 binding sequences (GAGGAA) located on antisense strand from position −1047 to −1042, −1098 to −1093, and −1423 to −1418 relative to the transcriptional start site of the *tp53* promoter (*Figure 5—figure supplement 2*). These observations indicate that Pu.1 might regulate the activity of P53 through direct protein-protein or protein-DNA interactions. Previously, we have shown that the level of *csf1ra*, the zebrafish orthologue of mammalian *Csf1r*, is highly associated with the fitness of microglia and regulates the turnover and maintenance of microglia via cell competition (*Yu et al., 2023*). During the process of investigation, we also wondered whether the cell competition between *pu.1*-deficient microglia and *pu.1*-sufficient microglia relied on Csf1ra. However, in contrast to the robust upregulation of *p53* in *pu.1*-deficient microglia, the decrease of *csf1ra* was not observed (*Figure 5—figure supplement 3*). In summary, our findings suggest that the dosage of PU.1/Pu.1 may serve as a checkpoint to determine the fitness of microglia. It will be of great interest to explore the upstream events that modulate *SPI1*/*pu.1* expression during microglia turnover.

## Materials and methods
### Zebrafish
Zebrafish were maintained according to standard protocol as described previously (*Westerfield, 2000*). AB wild-type, *pu.1*$^{Δ839}$ (*Yu et al., 2017*), *spi-b*$^{Δ232}$ (*Yu et al., 2017*), *tp53*$^{M214K}$ (*Berghmans et al., 2005*), *pu.1*$^{KI}$ (short for *pu.1*$^{KI}$-eGFP, shz4Tg), *pu.1*$^{CKO}$ (short for *pu.1*$^{KI}$-DsRed, shz5Tg), TgBAC(c-cl34b.1:eGFP) (*Wu et al., 2020*),Tg(coro1a:CreER$^{T2}$)$^{sz101tg}$, Tg(mpeg1:loxP-DsRedx-loxP-eGFP, shorted as LRLG)$^{hkz015t}$ (*Xu et al., 2016*) were used in this study.

### Mouse strains
*Spi1*$^{Fl}$ (No.T010980, purchased from GemPharmatech Co., Ltd) and *Cx3cr1*$^{CreER-EYFP}$ (Stock No. 021160) (*Yona et al., 2013*) were used in this study. All animal experiments and procedures were approved by the Ethics Committee for the Welfare of Laboratory Animals in Shenzhen PKU-HKUST Medical Center (reference number for approval: 2022–1166).

## Generation of *pu.1*<sup>KI</sup> allele

Donor plasmid was modified based on the reported Th-KI plasmid (*Li et al., 2015*). In brief, DNA fragments containing *pu.1* exon 4 to exon 6 and the 3' downstream fragment were cloned from cDNA and genomic DNA libraries, respectively, and inserted into the plasmid containing P2A-eGFP, P2A-Dsred and loxP elements. sgRNA was designed in the intron 3~4 of *pu.1* with the online website CRISPRscan (https://www.crisprscan.org/). It was transcribed by the MEGAshortscript Kit (AM1354, Invitrogen) from PCR product which was amplified following reported protocol (*Vejnar et al., 2016*). Then, Cas9 mRNA (700 ng/µl), sgRNA (70 ng/µl), and donor plasmid (15 ng/µl) were mixed and injected into zebrafish embryos at one-cell stage. Embryos with eGFP signal were selected at 3 dpf and raised to adulthood for germline transmission. Adult F0 fish were crossed with wild-type fish to generate F1 and eGFP⁺ F1 were selected. The 5'- and 3'-junctions of the donor plasmid integration site were amplified from the genomic DNA of F1 embryos and sent for sequencing. The primers used for 5' junction are 5'-GATCTATCGACCACCAATGGAG-3' and 5'-GCCATAGTGTGCATTCTCAGG-3' and for 3' junction are 5'-GTTGTAAAACGACGGCCAG-3' and 5'-GAGTGTAGTGCTCATTCAAGC-3'.

## Cryo-section

Adult fish were anesthetized on ice. The tissue was dissected and fixed in 4% PFA at 4°C overnight. After being washed with phosphate-buffered saline (PBS) at room temperature for 1 day, the tissues were dehydrated with 30% sucrose at 4°C overnight, then soaked in coagulating solution (optimal cutting temperature compound, OCT) and subjected to cryosection with 30 µm thickness.

## EdU and BrdU incorporation assays, TUNEL staining, and immunostaining

EdU and BrdU incorporation assays were conducted as previously reported (*Yu et al., 2023*). In brief, EdU (A10044, Invitrogen) or BrdU (B5002, Sigma-Aldrich) was dissolved in PBS to the final concentration of 10 mg/ml. 5 µl EdU or BrdU was intraperitoneally (i.p.) injected into adult zebrafish each time. Brain samples were dissected at 1 day post-injection and fixed in 4% PFA for cryosection.

EdU detection was conducted with (Click-iT EdU Cell Proliferation Kit for Imaging, Alexa Fluor 647 dye, Invitrogen, C10340) according to the manufacturer's instructions. For BrdU detection, samples were treated with 2 M HCl for half an hour and then stained with the BrdU antibody. TUNEL staining was performed with TUNEL BrightGreen Apoptosis Detection Kit (A112, Vazyme) according to the manufacturer's instructions. Antibody staining was performed after EdU/BrdU and TUNEL detection. Primary antibodies used in this study were mouse anti-BrdU (11170376001, Roche), goat anti-GFP (ab6658, Abcam), rabbit anti-DsRed (632496, Clontech), rabbit anti-Lcp1 (*Jin et al., 2009*), rabbit anti-Pu.1 (*Jin et al., 2012*) and rabbit anti-IBA1 (019–19741, FUJIFILM Wako) antibodies.

## Tamoxifen injection

4-Hydroxytamoxifen (4-OHT, H7904, Sigma) was dissolved in ethanol to final concentration of 10 mM. For adult fish, 1 µl 4-OHT was i.p. injected into each fish three times. For adult mice, tamoxifen (T5648, Sigma) was dissolved in corn oil to final concentration of 20 mg/ml and i.p. injected into the mice with 35 mg/kg (sub-dosage) or 150 mg/kg (overdosage).

## Neutral red staining

Neutral red staining was performed as previously described (*Herbomel et al., 2001*).

## Cy5-Annexin V treatment

Annexin V-Cy5 Reagent (1013, Biovision) solution was directly injected into brain ventricle of 3 dpf larva and immediately imaged by confocal microscope.

## Imaging

Fluorescent signals from live reporter lines or immunostaining were imaged under a Leica SP8 confocal microscope or a Zeiss LSM980 confocal microscope. The objectives HC PL APO 203/0.70 DRY (Leica) and Plan-Apochromat ×20/0.8 M27 (Zeiss) were used in this study.

## cDNA preparation and RNA-seq

Cell suspensions were prepared as previously described (*Yu et al., 2017*). Cells of interest were manually picked under the fluorescent microscope (Nikon Ti-S) equipped with the micro-manipulator (NT-88-V3, Nikon). Subsequently, picked cells were transferred by the glass needle to lysis buffer (mainly including Triton X-100 and RNase inhibitor) for thorough vortexing. Then the cell lysate was used for reverse transcription and amplification with the Smart-Seq2 method (*Picelli et al., 2014*) to generate the cDNA library. cDNA was sent to Novogene Company for Illumina sequencing with an average depth of $6\times10^6$ raw reads per sample. Reads were aligned to the Zebrafish Genome Assembly GRCz11 using the STAR (Spliced Transcripts Alignment to a Reference). Original counts were calculated with featureCounts package. DEGs were identified with the DESeq2 package. TPM (transcripts per million) are used to measure gene expression levels by normalization of counts. Samples with strong expression of microglia-related genes and faint DC-related genes were chosen for further analysis. TPM of top 50 up- and down-regulated genes were used to generate a heat map. TPM values in the heatmap were centered and scaled in the row direction. Noted that across all experimental groups subjected to RNA-seq analysis, the *ccl34b.1/ccl19a.1* expression ratios exceeded 5. Although residual DC contamination in the RNA-seq data cannot be entirely ruled out, we believed that microglia were the major population we analyzed in the RNA-seq.

## Mouse microglia isolation

Adult mice brain was dissected and cut into small pieces. They were dissociated into single-cell suspensions with Adult Brain Dissociation Kit (130-107-677, Miltenyi Biotec) according to the manufacturer's instructions. Then, Myelin Removal Beads II (130-96-733, Miltenyi Biotec) were used to remove myelin debris from single-cell suspensions. YFP+ microglia were sorted from suspensions by FACS Aria IIIu. The sorted microglia were lysed by proteinase K to extract DNA or TRIzol to extract RNA.

## TA clone assay

*Spi1Fl* and *Spi1KO* alleles were amplified by PCR from the genomic DNA. The common forward primer was 5'-GCATCGCATTGTCTGAGTAGGT-3. The reverse primers were 5'-AAATCTGCCTGGGTGA CCTTC-3' for *Spi1Fl* and 5'-ACACAACGGGTTCTTCTGTTAG-3' for *Spi1KO*. The PCR products for *Spi1Fl* and *Spi1KO* alleles were 248 bp and 129 bp, respectively. PCR products were ligated to pUCm-T vector (B522211, Sangon) with Blunt/TA Ligase Master Mix (M0367S, NEB) according to the manufacturer's instructions. The clones containing ligated plasmids were amplified with M13F (5'-GTTGTAAAACGA CGGCCAG-3') and M13R (5'-CAGGAAACAGCTATGAC-3'). The PCR products for *Spi1Fl* and *Spi1KO* alleles were 437 bp and 315 bp, respectively. Clones of *Spi1Fl* and *Spi1KO* alleles were distinguished by the size of PCR products.

## Quantitative PCR

cDNA was synthesized from extracted RNA with SuperScript III Reverse Transcriptase kit (18080093, Invitrogen). Quantitative PCR was performed with iTaq Universal SYBR Green Supermix (1725121, Bio-Rad) on a CFX96 Dx Instrument (Bio-Rad) and analyzed using the ΔΔCt method. The following primers were used to determine the expression of *DsRed* and *pu.1* from *pu.1CKO* allele: *DsRed*, 5'-GATCTATC GACCACCAATGGAG-3' (*pu.1*-161FP)/ 5'-GCCGTTCACGGAGCCCTCCAT-3' (DsRed-72RP); *pu.1*, 5'-GATCTATCGACCACCAATGGAG-3' (*pu.1*-161FP)/ 5'-CGCATGTAGTGACTGCACGC-3' (*pu.1*-357RP). The following primers were used to determine the expression of *Spi1* and *Spi-b* in mouse microglia: *Gapdh*, 5'-TGTGTCCGTCGTGGATCTGA-3'/5'-TTGCTGTTGAAGTCGCAGGAG-3'; *Spi1*, 5'- GAGG TGTCTGATGGAGAAGCTG-3'/5'-ACCCACCAGATGCTGTCCTTCA-3'; *Spi-b*, 5'- AGGAGTCTTCTA CGACCTGGAC-3'/5'-GGAGTGGCTAAAGGCAGCAGTA-3'.

## Quantification and statistics

The number of microglia in embryos were quantified manually, and then genotyped to prevent bias. For the experiment with adults, fish were genotyped first and then numbered without gender bias for downstream 4-OHT injection, brain sample collection, cryo-section, and immunostaining experiments. For microglia number quantification, 3 representative slices from anterior, middle and posterior of the midbrain of numbered fish were chosen for further manual counting with the same criteria. For EdU/ BrdU or TUNEL experiments, at least six representative slices from anterior, middle and posterior of

the midbrain were chosen for calculation. The number of EdU/BrdU$^+$ or TUNEL$^+$ microglia/DCs from the slices were counted and then divided by the total number of microglia/DCs to calculate the ratio.

Adult mice were genotyped first and then only males were chosen for the downstream experiments. Female mice were excluded to avoid variability associated with estrous cycle-dependent hormonal changes, which are known to influence microglial behavior (*Habib and Beyer, 2015*). For statistics, cortex, hippocampus, and thalamus on the slices of mouse brain were chosen for microglia density quantification. The number of microglia and region of interest were calculated with Imaris 9.9 software to determine the density.

Statistical analyses are performed with GraphPad Prism 9.5.1. F-test was first used to check the variances of two groups. If the variances of two groups were not significantly different ($p > 0.05$), Student's t-test was performed. If the variances of the two groups were significantly different ($p < 0.05$), t-test with Welch's correction was performed. A result was considered significant if $p < 0.05$. Values represent mean ± standard deviation of the mean.

## Acknowledgements

We thank Dr. Keng Chen and Dr. Keyu Chen in Shenzhen Bay Laboratory for their assistance in the Fluorescent-activated cell sorting (FACS) of mouse microglia, and thank Prof. Jin Xu in South China University of Technology for his kind suggestions on this project. This work is supported by National Natural Science Foundation of China Grant (31801211), Natural Science Foundation of Guangdong Province Grant (2024A1515030122), Shenzhen Medical Academy of Research and Translation (SMART) 2023 Shenzhen Medical Research Funding (B2302034), and Research Grants Council of the Hong Kong Special Administrative Region Grants (16103920, AoE/M-09/12, T13-605/18 W, and T13-602/21 N).

## Additional information

### Funding

| Funder | Grant reference number | Author |
|---|---|---|
| National Natural Science Foundation of China | 31801211 | Tao Yu |
| National Natural Science Foundation of Guangdong Province | 2024A1515030122 | Tao Yu |
| Shenzhen Medical Academy of Research and Translation | B2302034 | Zilong Wen Tao Yu |
| Research Grants Council, University Grants Committee | 16103920 | Zilong Wen |
| Research Grants Council, University Grants Committee | AoE/M-09/12 | Zilong Wen |
| Research Grants Council, University Grants Committee | T13-605/18-W | Zilong Wen |
| Research Grants Council, University Grants Committee | T13-602/21-N | Zilong Wen |

The funders had no role in study design, data collection and interpretation, or the decision to submit the work for publication.

### Author contributions

Yi Wu, Data curation, Formal analysis, Investigation, Visualization, Methodology, Writing – original draft, Writing – review and editing; Weilin Guo, Data curation, Formal analysis, Investigation,

Visualization, Methodology; Haoyue Kuang, Xiaohai Wu, Thi Huong Trinh, Yuexin Wang, Investigation; Shizheng Zhao, Formal analysis; Zilong Wen, Conceptualization, Funding acquisition, Writing – review and editing; Tao Yu, Conceptualization, Data curation, Formal analysis, Supervision, Funding acquisition, Visualization, Methodology, Writing – original draft, Project administration, Writing – review and editing

## Author ORCIDs
Yi Wu ⓘ https://orcid.org/0000-0002-9907-9403
Weilin Guo ⓘ http://orcid.org/0000-0001-6620-2762
Haoyue Kuang ⓘ http://orcid.org/0000-0001-6843-1788
Xiaohai Wu ⓘ https://orcid.org/0000-0002-2569-9239
Thi Huong Trinh ⓘ https://orcid.org/0000-0002-8799-8760
Shizheng Zhao ⓘ http://orcid.org/0000-0002-9859-7559
Zilong Wen ⓘ https://orcid.org/0000-0002-4260-7682
Tao Yu ⓘ https://orcid.org/0000-0001-6017-4852

## Ethics
All animal experiments and procedures were approved by the Ethics Committee for the Welfare of Laboratory Animals in Shenzhen PKU-HKUST Medical Center (reference number for approval: 2022-1166).

Reviewer #1 (Public review): https://doi.org/10.7554/eLife.105788.3.sa1
Reviewer #2 (Public review): https://doi.org/10.7554/eLife.105788.3.sa2
Author response https://doi.org/10.7554/eLife.105788.3.sa3

# Additional files

## Supplementary files
Supplementary file 1. The table generated by Deseq2 shows read counts, fold changes, and adjusted p-values of RNA-seq results.

## Data availability
Sequencing data have been deposited in GEO under accession code GSE283170. A table generated by Deseq2 has been included as a supplemental file to show read counts, fold changes, and adjusted p- values (*Supplementary file 1*).

The following dataset was generated:

| Author(s) | Year | Dataset title | Dataset URL | Database and Identifier |
| --- | --- | --- | --- | --- |
| Wu Y, Guo W, Kuang H, Wu X, Trinh TH, Wang Y, Zhao S, Wen Z, Yu T | 2024 | Effects of long-term depletion of Pu.1 in adult zebrafish microglia in mosaic condition | https://www.ncbi.nlm.nih.gov/geo/query/acc.cgi?acc=GSE283170 | NCBI Gene Expression Omnibus, GSE283170 |

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
