## [Editor Report · eLife Assessment]

This study presents **valuable** findings on the regulation of survival and maintenance of brain-resident immune cells called microglia. Using **compelling** and sophisticated genetic tools, the authors demonstrate a gene dosage-dependent mechanism using which microglia are eliminated. This research on cell competition and survival will be of broad interest to the cell biology community.

---

## [Referee Report · Reviewer #1 (Public review)]

Summary:

The article entitled "Pu.1/Spi1 dosage controls the turnover and maintenance of microglia in zebrafish and mammals" by Wu et al., identifies a role for the master myeloid developmental regulator Pu.1 in the maintenance of microglial populations in the adult. Using a non-homologous end joining knock-in strategy, the authors generated a pu.1 conditional allele in zebrafish, which reports wildtype expression of pu.1 with EGFP and truncated expression of pu.1 with DsRed after Cre mediated recombination. When crossed to existing pu.1 and spi-b mutants, this approach allowed the authors to target a single allele for recombination and induce homozygous loss-of-function microglia in adults. This identified that although there is no short-term consequence to loss of pu.1, microglia lacking any functional copy of pu.1 are depleted over the course of months, even when spi-b is fully functional. The authors go on to identify reduced proliferation, increased cell death, and higher expression of tp53 in the pu.1 deficient microglia, as compared to the wildtype EGFP+ microglia. To extend these findings to mammals, the authors generated a conditional Pu.1 allele in mice and performed similar analyses, finding that loss of a single copy of Pu.1 resulted in similar long-term loss of Pu.1-deficient microglia. The conclusions of this paper are overall well supported by the data.

Strengths:

The genetic approaches here for visualizing recombination status of an endogenous allele are very clever, and by comparing the turnover of wildtype and mutant cells in the same animal the authors can make very convincing arguments about the effect of chronic loss of pu.1. Likely this phenotype would be either very subtle or non-existent without the point of comparison and competition with the wildtype cells.

Using multiple species allows for more generalizable results, and shows conservation of the phenomena at play.

The demonstration of changes to proliferation and cell death in concert with higher expression of tp53 is compelling evidence for the authors argument.

Weaknesses:

This paper is very strong. It would benefit from further investigating the specific relationship between pu.1 and tp53 specifically. Does pu.1 interact with the tp53 locus? Specific molecular analysis of this interaction would strengthen the mechanistic findings.

Recommendations for the authors It would be useful to investigate the relationship between pu.1 and tp53. The data presented here show that pu.1 deficient cells have higher expression of tp53, but this could be an indirect effect. However, since pu.1 has known DNA binding motifs, it would be worthwhile to investigate if there are any direct interactions between pu.1 and the tp53 locus -- does pu.1 directly bind and repress tp53 expression? This could be directly investigated with Cut & Run or an EMSA.

The paper would likely also benefit from more in-depth discussion of the relationship of the zebrafish alleles and their relationship to mammalian Pu.1 -- as presented here, the authors are implicitly arguing that zebrafish pu.1 and spi-b are both more closely related to mammalian Pu.1 than to mammalian Spi-b. Clear argument, perhaps backed up by sequence alignment and homology matching, would help readers, especially those less familiar with zebrafish genome duplications.

Comments on Revised Version (from BRE):

The authors performed in silico analyses to support a regulatory relationship between Pu.1 and Tp53. They identified three putative Pu.1 binding sites within the zebrafish tp53 promoter region. Furthermore, they cite prior evidence demonstrating a similar interaction between PU.1 and members of the P53 family through direct DNA binding.

---

## [Referee Report · Reviewer #2 (Public review)]

Summary:

In the presented work by Wu et al. the authors investigate the role of the transcription factor Pu.1 in the survival and maintenance of microglia, the tissue resident macrophage population in the brain. To this end they generated a sophisticated new conditional pu.1 allele in zebrafish using CRISPR mediated genome editing which allows visual detection of expression of the mutant allele through a switch from GFP to dsRed after Cre-mediated recombination. Using EdU pulse-chase labelling, they first estimate the daily turnover rate of microglia in the adult zebrafish brain which was found to be higher than rates previously estimated for mice and humans. After conditional deletion of pu.1 in coro1a positive cells, they do not find a difference in microglia number at 2 and 8 days or 1 month post injection of Tamoxifen. However, at 3 month post injection, a strong decrease in mutant microglia could be detected. While no change in microglia number was detected at 1mpi, an increase in apoptotic cells and decreased proliferation as observed. RNA-seq analysis of WT and mutant microglia revealed an upregulation of tp53, which was shown to play a role in the depletion of pu.1 mutant microglia as deletion in tp53-/- mutants did not lead to a decrease in microglia number at 3mpi. Through analysis of microglia number in pU.1 mutants, the authors further show that the depletion of microglia in the conditional mutants is dependent on the presence of WT microglia. To show that the phenomenon is conserved between species, similar experiments were also performed in mice.

This work expands on previous in vitro studies using primary human microglia. The majority of conclusions are well supported by the data, addition of controls and experimental details would strengthen the conclusions and rigor of the paper.

Strengths:

Generation of an elegantly designed conditional pu.1 allele in zebrafish that allows for the visual detection of expression of the knockout allele.

The combination of analysis of pu.1 function in two model systems, zebrafish and mouse, strengthens the conclusions of the paper.

Confirmation of the functional significance of the observed upregulation of tp53 in mutant microglia through double mutant analysis provides some mechanistic insight.

Weaknesses:

(1) The presented RNA-Seq analysis of mutant microglia is underpowered and details on how the data was analyzed is missing. Only 9-15 cells were analyzed in total (3 pools of 3-5 cells each). Further the variability in relative gene expression of ccl35b.1, which was used as a quality control and inclusion criterion to define pools consisting of microglia, is extremely high (between ~4 and ~1600, Fig. S7A).

(2) The authors conclude that the reduction of microglia observed in the adult brain after cKO of pu.1 in the spi-b mutant background is due to apoptosis (Lines 213-215). However, they only provide evidence of apoptosis in 3-5 dpf embryos, a stage at which loss of pu.1 alone does lead to a complete loss of microglia (Fig.2E). A control of pu.1 KI/d839 mutants treated with 4-OHT should be added to show that this effect is indeed dependent on the loss of spi-b. In addition, experiments should be performed to show apoptosis in the adult brain after cKO of pu.1 in spi-b mutants as there seems to be a difference in requirement of pu.1 in embryonic and adult stages.

Comments on Revised Version (from BRE):

The authors have elaborated on the details of the RNA-Seq procedure and clarified the distinct phenotypes observed with global versus condition pu.1 knockout. In addition, the authors' proposed collaborative relationship between Pu.1 and Spi-b has been expanded in the revised manuscript. The authors have addressed all the minor concerns raised by the reviewer.

---

## [Author Response]

The following is the authors’ response to the original reviews

**Reviewer #1 (Public review):**
Strengths:The genetic approaches here for visualizing the recombination status of an endogenous allele are very clever, and by comparing the turnover of wildtype and mutant cells in the same animal the authors can make very convincing arguments about the effect of chronic loss of pu.1. Likely this phenotype would be either very subtle or nonexistent without the point of comparison and competition with the wildtype cells.Using multiple species allows for more generalizable results, and shows conservation of the phenomena at play.The demonstration of changes to proliferation and cell death in concert with higher expression of tp53 is compelling evidence for the authors' argument.Weaknesses:This paper is very strong. It would benefit from further investigating the specific relationship between pu.1 and tp53 specifically. Does pu.1 interact with the tp53 locus? Specific molecular analysis of this interaction would strengthen the mechanistic findings.

We agree with the reviewer’s assessment regarding the significance of the relationship between PU.1 and TP53. To investigate the potential interaction between Pu.1 and Tp53 in zebrafish, we analyzed the promoter region of zebrafish *tp53*. Indeed, we found three PU.1 binding sites (GAGGAA) on *tp53* promoter, which locate on the antisense strand from position -1047 to -1042, -1098 to -1093 and -1423 to -1418 relative to the transcriptional start site (Figure 5-figure supplement 2). These potential Pu.1 binding sites indicate a direct interaction between Pu.1 and *tp53* locus. Furthermore, a previous study by Tschan et al. (2008) elucidated the mechanism by which PU.1 attenuates the transcriptional activity of the P53 tumor suppressor family through direct binding to the DNA-binding and/or oligomerization domains of p53/p73 proteins. We have also cited this study (Line 399-401) and included all above information in the discussion of the revised manuscript (Line 399-405).

**Reviewer #2 (Public review):**
Strengths:Generation of an elegantly designed conditional pu.1 allele in zebrafish that allows for the visual detection of expression of the knockout allele.The combination of analysis of pu.1 function in two model systems, zebrafish and mouse, strengthens the conclusions of the paper.Confirmation of the functional significance of the observed upregulation of tp53 in mutant microglia through double mutant analysis provides some mechanistic insight.Weaknesses:(1) The presented RNA-Seq analysis of mutant microglia is underpowered and details on how the data was analyzed are missing. Only 9-15 cells were analyzed in total (3 pools of 3-5 cells each). Further, the variability in relative gene expression of ccl35b.1, which was used as a quality control and inclusion criterion to define pools consisting of microglia, is extremely high (between ~4 and ~1600, Figure S7A).

We feel sorry for the unclearness of RNAseq procedures and have accordingly added the details about RNA-seq data analysis in the “Material and methods” section (Line 491-501). Briefly, reads were aligned to the zebrafish genome using the STAR package. Original counts were calculated with featureCounts package. Differential expression genes (DEGs) were identified with the DESeq2 package. Owing to the technical challenge of unambiguously distinguishing microglia from dendritic cells (DCs) in brain cell suspensions, we employed a strategy of isolating 3-5 cells per pool and quantifying the relative expression of the microglia-specific marker *ccl34b.1* normalized to the DC-specific marker *ccl19a.1*. This approach aimed to reduce DC contamination in downstream analyses. Across all experimental groups subjected to RNA-seq analysis, the *ccl34b.1*/*ccl19a.1* expression ratios exceeded 5, confirming microglia as the dominant cell population. Nonetheless, residual DC contamination in the RNA-seq data cannot be entirely ruled out. We have discussed this technical constraint in the revised manuscript to ensure methodological transparency (Line 498-501).

(2) The authors conclude that the reduction of microglia observed in the adult brain after cKO of pu.1 in the spi-b mutant background is due to apoptosis (Lines 213-215). However, they only provide evidence of apoptosis in 3-5 dpf embryos, a stage at which loss of pu.1 alone does lead to a complete loss of microglia (Figure 2E). A control of pu.1 KI/d839 mutants treated with 4-OHT should be added to show that this effect is indeed dependent on the loss of spi-b. In addition, experiments should be performed to show apoptosis in the adult brain after cKO of pu.1 in spi-b mutants as there seems to be a difference in the requirement of pu.1 in embryonic and adult stages.

We apologize for the omission of data regarding conditional *pu.1* knockout alone in the embryos in our manuscript, which may have led to ambiguity. We would like to clarify that conditional *pu.1* knockout alone at the embryonic stage does not induce microglial death (Figure 3-figure supplement 1). Microglial death occurs only in both embryonic and adult brains when Pu.1 is disrupted in the *spi-b* mutant background. The blebbing morphology of some microglia after *pu.1* conditional knockout in adult *spi-b* mutant indicated microglia undergo apoptosis at both embryonic and adult stages (Figure 3-figure supplement 3 and Figure 3-figure supplement 4). The reviewer’s concern likely arises from the distinct outcomes of global *pu.1* knockout (Fig. 2) versus conditional *pu.1* ablation (Figure 3-figure supplement 1). Global knockout eliminates microglia during early development due to Pu.1’s essential role in myeloid lineage specification. We have included this clarification in the revised manuscript (Line 208-211).

(3) The number of microglia after pu.1 knockout in zebrafish did only show a significant decrease 3 months after 4-OHT injection, whereas microglia were almost completely depleted already 7 days after injection in mice. This major difference is not discussed in the paper.

We propose that zebrafish Pu.1 and Spi-b function cooperatively to regulate microglial maintenance, analogous to the role of PU.1 alone in mice. This cooperative mechanism likely explains the observed difference in microglial depletion kinetics between zebrafish and mice following *pu.1* conditional knockout. Specifically, the compensatory activity of Spi-b in zebrafish may buffer the immediate loss of Pu.1, whereas in mice, the absence of *Spi-b* expression in microglia eliminates this redundancy, resulting in rapid microglial depletion. Furthermore, during evolution, *SPI-B* appears to have acquired lineage-specific roles, becoming absent in microglia. We have included the clarification in the revised manuscript (Line 302-305).

(4) Data is represented as mean +/-.SEM. Instead of SEM, standard deviation should be shown in all graphs to show the variability of the data. This is especially important for all graphs where individual data points are not shown. It should also be stated in the figure legend if SEM or SD is shown

We have represented our data as mean ± SD in the revised manuscript.

**Recommendations for the authors:**

**Reviewing Editor:**
To further strengthen the manuscript, we ask the authors to address the reviewers' comments through additional experiments where necessary. In cases where certain experiments may be challenging, we encourage the authors to address these concerns within the text, such as by referencing any prior evidence of pu.1 and tp53 interactions or incorporating in silico analyses that support such interaction.

As suggested, we have performed in-silico analysis of Pu.1 binding sites in zebrafish tp53 promoter and also cited previous paper showing how PU.1 attenuates the transcriptional activity of the P53 tumor suppressor family (Line 399-405).

**Reviewer #1 (Recommendations for the authors):**
It would be useful to investigate the relationship between pu.1 and tp53. The data presented here show that pu.1 deficient cells have higher expression of tp53, but this could be an indirect effect. However, since pu.1 has known DNA binding motifs, it would be worthwhile to investigate if there are any direct interactions between pu.1 and the tp53 locus -- does pu.1 directly bind and repress tp53 expression? This could be directly investigated with Cut & Run or an EMSA.

The interaction between Pu.1 and Tp53 has been discussed in the public review section.

The paper would likely also benefit from a more in-depth discussion of the relationship of the zebrafish alleles and their relationship to mammalian Pu.1 -- as presented here, the authors are implicitly arguing that zebrafish pu.1 and spi-b are both more closely related to mammalian Pu.1 than to mammalian Spi-b. A clear argument, perhaps backed up by sequence alignment and homology matching, would help readers, especially those less familiar with zebrafish genome duplications.

We have conducted detailed sequence alignment in our previous work (Yu et al., 2017, Blood) and found zebrafish Spi-b shares the highest similarity with the mammalian SPI-B among Ets family transcription factors in zebrafish. A unique P/S/T-rich region known to be essential for mammalian SPI-B transactivation activity is present in zebrafish Spi-b. Our data do not support the interpretation that *Spi-b* is more closely related to mammalian *Pu.1* than to *Spi-b*. Instead, functional compensation between *pu.1* and *spi-b* in microglia maintenance likely reflects their shared role as Ets-family transcriptional regulators, rather than ortholog-driven redundancy.

**Reviewer #2 (Recommendations for the authors):**
(1) The nomenclature of the genes in the SPI family in zebrafish is somewhat confusing as genes were renamed several times. It would make it easier for the reader to understand if in the abstract and the main text, spi-b would be referred to as the zebrafish orthologue of mouse SPI-B (as determined by the authors in previous work) rather than the paralogue of zebrafish pu.1. To clarify which genes were analyzed in both zebrafish and mouse, Gene accession numbers should be added.

Thanks for the recommendations. We have changed “the paralogue of zebrafish pu.1” to “the orthologue of mouse *Spi-b*” in the abstract (Line 22) and added gene accession numbers for both zebrafish and mouse gene (Line 105-106 and 301-302).

(2) Methods RNA-seq: Details on how the aligned reads were analyzed to detect differentially expressed genes are missing and should be added. In addition, a table with read counts, fold changes and adjusted p values should be added.

We have added details of RNA-seq analysis in the Material and Methods part (Line 491-501). A table generated by Deseq2 has been included as a supplemental file to show read counts, fold changes and adjusted p values (Supplemental file 1).

(3) Figure 2H: It would be helpful to the reader if the KO splicing would be shown in comparison to WT splicing.

Thank you for your suggestion. We have added the sequence result between exon 3 and exon 4 of *pu.1* from wildtype cDNA to show WT splicing in Figure 2H.

(4) Legend Figure 5C. Relative expression should be replaced with transcripts per million (TPM).

We have corrected it in the figure legend of Figure 5C (Line 786-787).

(5) In Figure S3. the label on the y-axis in panel B is not visible.

We apologize for the mistake during figures assembling. We have corrected it and now the y-axis is visible.

(6) In Figure S7B an explanation for the colors in the heat map is missing and should be added.

Colors represent scaled TPM values. The red color represents high expression while the blue color represents low expression. We have added the information in the figure legend.

(7) A justification for the use of male mice only should be added or additional experiments in female mice should be performed.

Female mice were excluded to avoid variability associated with estrous cycle-dependent hormonal changes, which are known to influence microglial behavior (Habib P et al., 2015). We have added a justification in the revised manuscript (Line 547-548).

(8) The manuscript would benefit from some language editing. A few examples are listed below:a) line 97: the rostral blood (RBI) should read the rostral blood island.b) line 373 typo: nucleus translocation should read nuclear translocation.c) line 393 typo: pu.1-dificent should read pu.1-deficient.

We apologize for the typos or grammar mistakes in the manuscript. We have checked the manuscript thoroughly and revised those typos or grammar mistakes.

Reference:

Tschan MP, Reddy VA, Ress A, Arvidsson G, Fey MF, Torbett BE (2008) PU.1 binding to the p53 family of tumor suppressors impairs their transcriptional activity. Oncogene 27: 3489-93

Yu T, Guo W, Tian Y, Xu J, Chen J, Li L, Wen Z (2017) Distinct regulatory networks control the development of macrophages of different origins in zebrafish. Blood 129: 509-519

Habib P, Beyer C (2015) Regulation of brain microglia by female gonadal steroids. J Steroid Biochem Mol Biol 146: 3-14